# Cryo-EM structure of human Pol κ bound to DNA and mono-ubiquitylated PCNA

Claudia Lancey [1,4], Muhammad Tehseen[2,4], Souvika Bakshi [1], Matthew Percival[1], Masateru Takahashi[2], Mohamed A. Sobhy[2], Vlad S. Raducanu [2], Kerry Blair[1], Frederick W. Muskett [1], Timothy J. Ragan [1], Ramon Crehuet [3], Samir M. Hamdan [2✉] & Alfredo De Biasio[1,2✉]

Y-family DNA polymerase κ (Pol κ) can replicate damaged DNA templates to rescue stalled replication forks. Access of Pol κ to DNA damage sites is facilitated by its interaction with the processivity clamp PCNA and is regulated by PCNA mono-ubiquitylation. Here, we present cryo-EM reconstructions of human Pol κ bound to DNA, an incoming nucleotide, and wild type or mono-ubiquitylated PCNA (Ub-PCNA). In both reconstructions, the internal PIP-box adjacent to the Pol κ Polymerase-Associated Domain (PAD) docks the catalytic core to one PCNA protomer in an angled orientation, bending the DNA exiting the Pol κ active site through PCNA, while Pol κ C-terminal domain containing two Ubiquitin Binding Zinc Fingers (UBZs) is invisible, in agreement with disorder predictions. The ubiquitin moieties are partly flexible and extend radially away from PCNA, with the ubiquitin at the Pol κ-bound protomer appearing more rigid. Activity assays suggest that, when the internal PIP-box interaction is lost, Pol κ is retained on DNA by a secondary interaction between the UBZs and the ubiquitins flexibly conjugated to PCNA. Our data provide a structural basis for the recruitment of a Y-family TLS polymerase to sites of DNA damage.

[1] Leicester Institute of Structural & Chemical Biology and Department of Molecular & Cell Biology, University of Leicester, Lancaster Rd, Leicester LE1 7HB, UK. [2] Division of Biological and Environmental Sciences and Engineering, King Abdullah University of Science and Technology, Thuwal 23955, Saudi Arabia. [3] CSIC-Institute for Advanced Chemistry of Catalonia (IQAC) C/ Jordi Girona 18-26, 08034 Barcelona, Spain. [4] These authors contributed equally: Claudia Lancey, Muhammad Tehseen. ✉email: samir.hamdan@kaust.edu.sa; alfredo.debiasio@kaust.edu.sa

Cells are continuously subjected to DNA damage caused by environmental mutagens and reactive metabolites, which threaten the stability of the cell genome[1,2]. At a DNA lesion, the cell faces a choice between stalling DNA replication or employing a more error-prone replication system that tolerates the damage before it can be repaired. Translesion DNA synthesis (TLS) is the process that allows cells to overcome the deleterious effects of replication stalling and genomic instability caused by DNA damage[3–6]. While being of the utmost importance for cell survival, TLS is also intrinsically mutagenic and is implicated in human cancer[7–9]. Eukaryotic TLS involves canonical high-fidelity as well as specialized error-prone TLS polymerases (e.g., Y-family Pol η, Pol ι, Pol κ and Rev1), which can synthesize DNA past a lesion due to their active sites being able to accommodate damaged templates[3,10,11]. Both high-fidelity and TLS polymerases form complexes with the homotrimeric sliding clamp proliferating cell nuclear antigen (PCNA), which encircles duplex DNA and tethers these polymerases to the template, enhancing their processivity[12–14]. At the lesion site, the high-fidelity polymerase stalls and PCNA is mono-ubiquitylated at K164 by the Rad6–Rad18 ubiquitin ligase complex[15–17]; a TLS polymerase then binds to the resident PCNA and replicates the damaged DNA[18]. PCNA ubiquitylation facilitates the recruitment and retention of TLS polymerases to the damage sites in vivo[19–25] and in a fully reconstituted yeast replisome[26]. Y-family TLS polymerases possess both PCNA-interacting motifs (PIP-boxes) and ubiquitin binding modules, which are believed to interact with the hydrophobic cleft on the front face of PCNA and with the ubiquitin moieties covalently attached to PCNA, respectively[13]. However, the structural basis of the interaction of TLS polymerases with both DNA and unmodified or mono-ubiquitylated PCNA, and therefore the mechanism of TLS polymerase recruitment to sites of DNA damage, remain poorly understood.

Pol κ can bypass several types of DNA lesions, mainly at the $N^2$ position of guanine in an error-free manner[27], and efficiently extend mispaired termini with lower misincorporation frequency than undamaged templates[28]. Orthologs of Pol κ exist in bacteria and archaea, including DinB (Pol IV) in *Escherichia coli* and Dbh and Dpo4 in *Sufolobus solfataricus*[27]. Pol κ shares a similar domain architecture with Pol η and ι, consisting of an N-terminal catalytic core (comprising a palm, fingers, thumb, and polymerase-associated domain (PAD)) and a long C-terminal domain containing two PIP-boxes, one Rev-1 interacting motif (RIR) and two Ubiquitin Binding Zinc Fingers (UBZs), and predicted to be largely unstructured[29] (Fig. 1a). An extension of ~75 amino acids at the N-terminus (N-clasp), which is functionally important, is a unique feature of Pol κ[30]. The structure of the catalytic domain of human Pol κ has been solved in the apo form and in complex with DNA[30,31]. The apo and DNA-bound structures of Pol κ display a large difference in the orientation of the PAD relative to the thumb domain. The apo-enzyme was crystallized with the PAD positioned under the palm domain in two alternate positions, while in the DNA-bound structure the PAD is docked in the major groove; for the most divergent position, a movement requiring a 50 Å shift and a 143° rotation[30] (Fig. 1b). Conformational freedom of the PAD in the apo form has also been observed in Dpo4 [32] and, to a minor extent, in Pol η[33,34], and seems to be a general feature of Y-family polymerases. DNA binding to Pol κ also results in the folding of the N-clasp into two helices (αN1 and αN2) encircling the primer-template junction[30].

In this work, we present cryo-EM reconstructions of full-length human Pol κ bound to a primer/template (P/T) DNA substrate, an incoming nucleotide, and wild-type PCNA (wt-PCNA) or PCNA mono-ubiquitylated at K164 (Ub-PCNA), at resolutions between ~3.4 and ~6.4 Å. The structures of the complexes with either wt-PCNA or Ub-PCNA are very similar and show the Pol κ core docked to one PCNA protomer through the internal PIP-box adjacent to the PAD. The region C-terminal to the internal PIP-box, containing the UBZs, is instead invisible in the cryo-EM maps, suggesting it is flexible. The complexes' architecture is unusual, displaying the catalytic domain and the DNA exiting the Pol κ core sharply angled relative to the PCNA ring. Our MD simulations predict that, in the absence of DNA, Pol κ bound to PCNA is conformationally flexible, implying that binding to DNA is required for the assembly of the rigid and active holoenzyme. The cryo-EM reconstruction of Pol κ complexed with Ub-PCNA displays weak density protruding radially away from the back face of PCNA where the three lysine residues at position 164 are located, suggesting partial flexibility of the ubiquitin moieties; density for the ubiquitin attached to the PCNA protomer bound to Pol κ is better defined and is consistent with ubiquitin orientations having the hydrophobic surface exposed for potential interaction with Pol κ UBZs. In agreement, primer extension assays with Pol κ variants containing mutations in the internal PIP-box and UBZs show that Pol κ activity depends primarily on the internal PIP-box interaction and, when this interaction is lost, on a secondary interaction between Pol κ UBZs and the ubiquitins flexibly attached to Ub-PCNA. Collectively, these data provide a structural framework to explain how PCNA recruits a Y-family TLS polymerase to sites of DNA damage.

## Results

**Architecture of Pol κ bound to DNA and wild-type PCNA.** We reconstituted the Pol κ−DNA−wt-PCNA complex by mixing purified recombinant Pol κ, PCNA, a (25/38) P/T DNA substrate containing a dideoxy chain terminator in the primer strand, and dTTP as the incoming nucleotide. The complex was separated by micro-size exclusion chromatography (Supplementary Fig. 1), vitrified and imaged by cryo-EM (Supplementary Fig. 2). We obtained a reconstruction of the complex at a global resolution of 3.4 Å (Fig. 1c−e, Supplementary Figs. 2–3 and Supplementary Table 1). The structure has approximate dimensions of $127.7 \times 88.0 \times 89.1$ Å, and displays the catalytic core of Pol κ sitting on top of the front face of PCNA in a remarkably angled orientation, with the axis of DNA in the catalytic cleft tilted by ~47° relative to the normal of the PCNA ring plane (Fig. 1f). The duplex DNA emerging from the catalytic core threads through the PCNA ring hole and bends by ~30° to avoid clashing with the ring inner rim (Fig. 1f). The long region C-terminal to the PAD (residues 535−870, Fig. 1a) is invisible in the map, suggesting it is flexible, in agreement with the disorder prediction (Fig. 1a). Fitting of the X-ray structure of Pol κ catalytic domain bound to DNA (PDB ID 2OH2)[30] into the cryo-EM map shows an excellent correlation for Pol κ palm, fingers, thumb, PAD and P/T DNA in the active site, while bending of dsDNA in the Pol κ holoenzyme results in the poor fitting of the bases below the PAD. The map quality allowed us to build an atomic model of the visible part of the complex (Fig. 1e). Residues 21−45 in the N-clasp are invisible in the map (Supplementary Fig. 4), suggesting that helix αN1 in the N-clasp can be partly flexible even in the presence of DNA. This agrees with the high average B-factor of residues 21−44 in the X-ray structure (107.1 Å²) compared to the global average value (69.5 Å²), and with the fact that αN1 in the N-Clasp engages in marginal interactions with DNA[30]. In addition, the weak cryo-EM density of Pol κ thumb helix αA adjacent to N-clasp α2 (Supplementary Fig. 4) suggests that, in the absence of stabilizing crystal contacts, helix αA is quite dynamic.

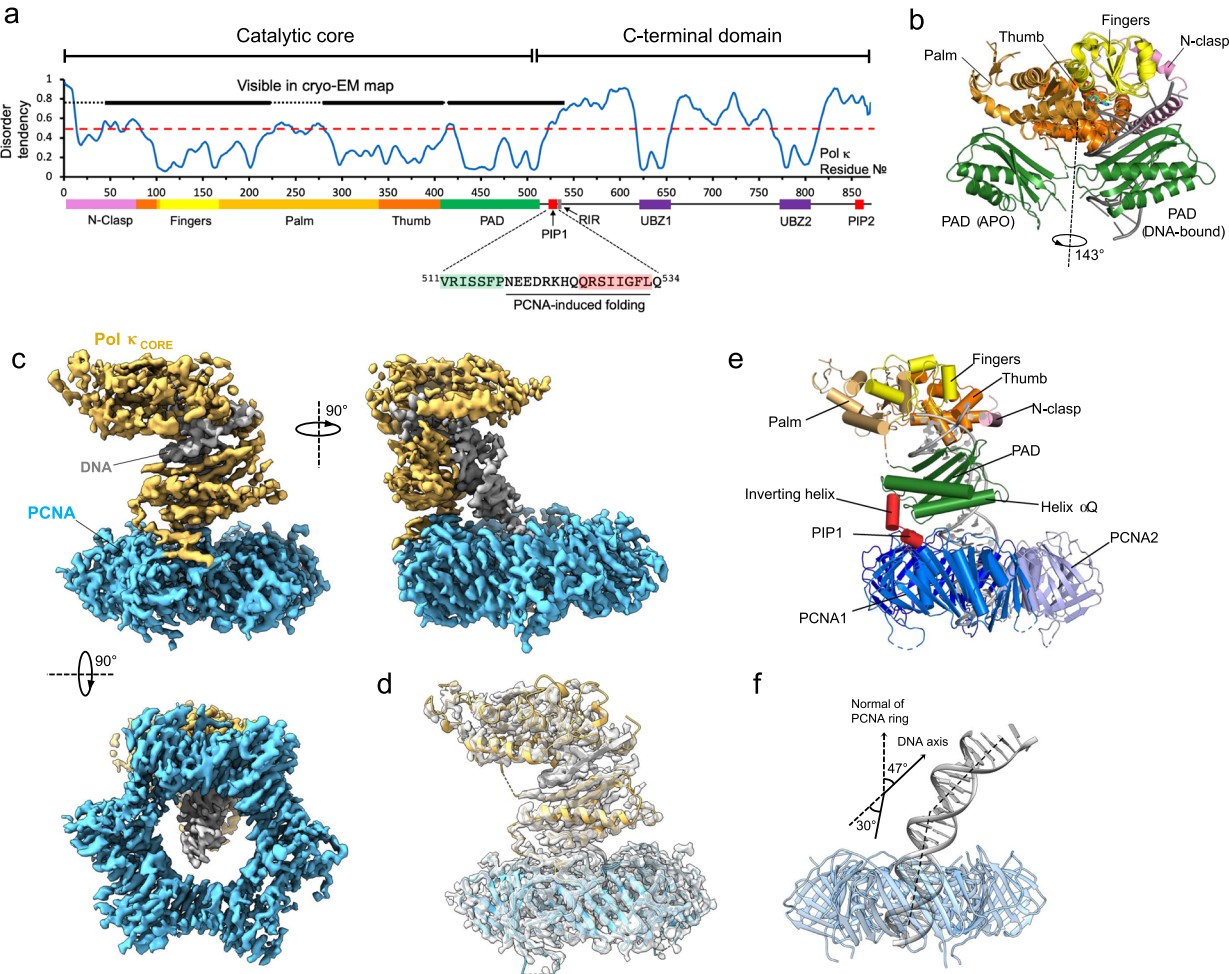

**Fig. 1 Cryo-EM structure of the Pol κ−DNA−wt-PCNA complex. a** Disorder prediction against residue number of human Pol κ, and Pol κ domain organization and amino acid sequence of the PCNA-interacting region; PIP PCNA-interacting motif, RIR Rev-1 interacting motif, UBZ Ubiquitin binding zinc-finger. Disorder prediction was performed with PrDOS[94]. The red dotted line corresponds to a disorder tendency of 0.5. The thick black line corresponds to Pol κ residues observed in the cryo-EM reconstruction. **b** X-ray structures of apo- (PDB ID 1T94)[31] and DNA-bound Pol κ (PDB ID 2OH2)[30] overlaid on the core domain. The PAD in the apo structure is rotated 143° relative to the PAD in the DNA-bound structure. Pol κ sub-domains are colored as in panel (**a**). **c** Cryo-EM density map of Pol κ complex colored by components (Pol κ in orange, PCNA in skyblue and DNA in gray). **d** Structural model fitted into the cryo-EM map. **e** Structure of Pol κ complex colored by domain. **f** DNA bending in Pol κ holoenzyme model. PCNA is shown as a transparent blue ribbon, DNA as a gray ribbon. Pol κ core and PAD were removed for clarity.

**Interaction of Pol κ with PCNA.** The map resolution at the Pol κ−PCNA interface (~3.2 Å; Supplementary Fig. 2) was sufficient for the de novo model building of this region (Figs. 1e, 2a, b). Pol κ interacts with one of the three PCNA protomers through the C-terminal region of the PAD spanning residues 517−534, which is disordered in the absence of PCNA[30] and becomes structured in the complex. Specifically, residues 518−525 fold into a two-turn α-helix ("Inverting helix", Figs. 1e, 2b), which reverses the chain direction and inserts the PIP-box (526QRSIIGFL) between the loop connecting helix αQ and β11 of the PAD, the hydrophobic cleft on the front face of PCNA, and the PCNA C-terminus (Fig. 2a, b). The PIP-box acquires the canonical 3.10 helix conformation and docks to the PCNA groove via a three-fork plug made of side chains of Ile529, Phe532 and Leu533, while Gln526 binds in the so-called "Q-pocket" (Fig. 2a, b). While deviating from the strict PIP-box consensus sequence (Qxxhxxaa, where h is a hydrophobic, a is an aromatic, and x is any residue), the Pol κ PIP-box interacts through the prototypical molecular surface observed in other PCNA-interacting partners[35]. Additional interactions further stabilize the structure: two main-chain hydrogen bonds between Gln525 and Arg527 of Pol κ and Ile255

and Pro253 in the C-terminus of PCNA, and two hydrogen bonds between His44 on a PCNA loop adjacent to the hydrophobic cleft and residues Ser528 and Ile529 within the Pol κ PIP-box (Fig. 2b). Thus, the folding and concomitant insertion of the PAD C-terminus between the PAD and PCNA creates a composite interface burying a total of 733 Å², and bends Pol κ core over the bound PCNA protomer, causing the bending of DNA threading the PCNA pore. Interestingly, both Pol κ and Pol δ interact with only one PCNA protomer via a PIP-box interaction involving the C-terminus of the catalytic domain, but the polymerase chains approach the PCNA binding groove from opposite directions and are connected N-terminally to distinct domains (Fig. 2c).

**Interactions of DNA with Pol κ and PCNA.** In the active site of Pol κ, Watson−Crick base pairing is observed between the incoming dTTP and the opposing A in the template strand (Fig. 2a, d). The triphosphate of dTTP inserts between the palm and fingers domain and its position is locked by hydrogen bonding with Tyr111, Arg144 and Lys328, three conserved residues among Y-family polymerases (Fig. 2d). The side chains of

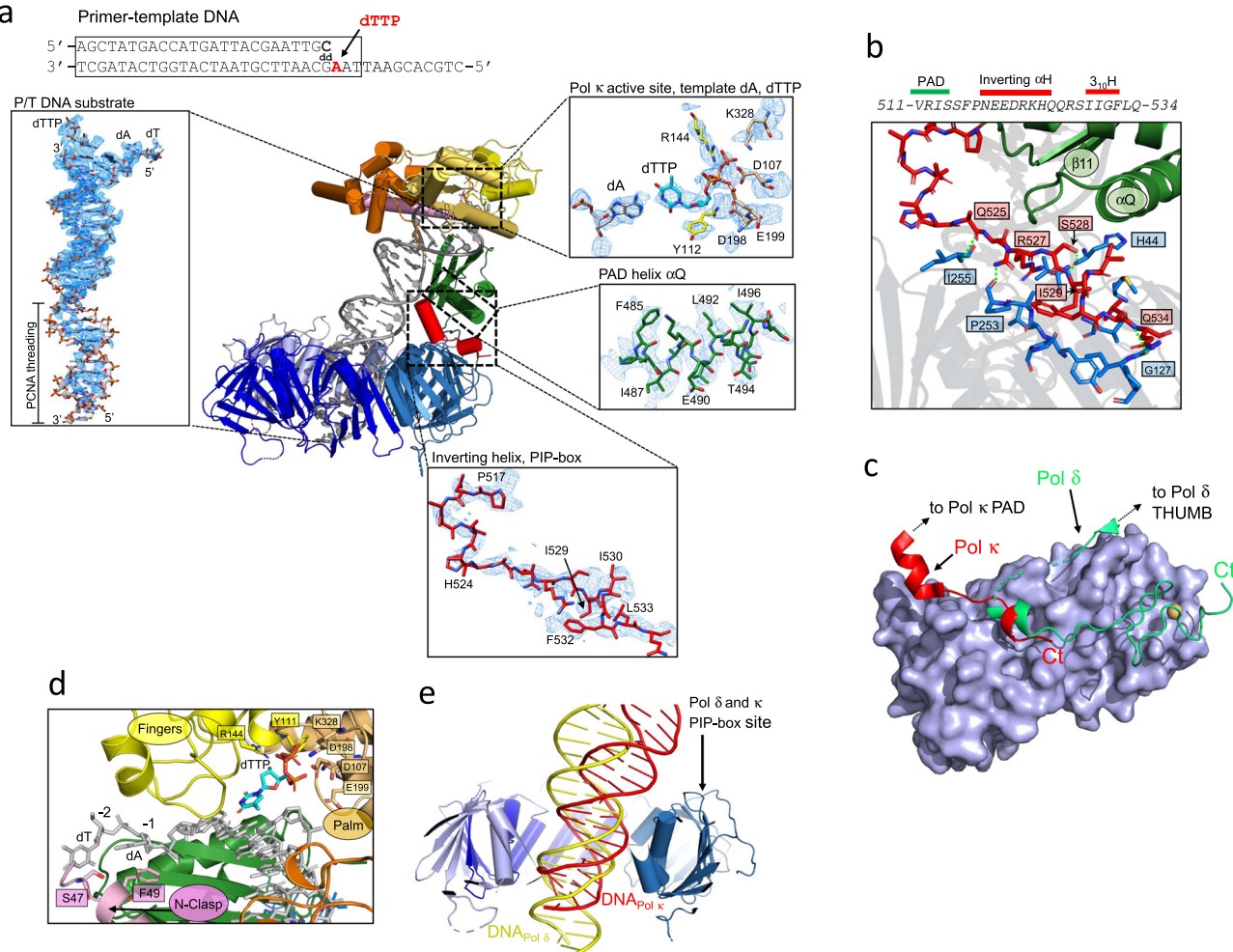

**Fig. 2 Details of the cryo-EM structure of the Pol κ−DNA−wt-PCNA complex and comparison with Pol δ processive holoenzyme structure. a** Map region around different elements of the Pol κ−DNA−wt-PCNA complex. The sequence of the DNA P/T substrate is shown. The region of the substrate that was modeled is boxed. **b** Inter-molecular interactions at the Pol κ−PCNA binding site. Pol κ PIP-box and PCNA-interacting residues are shown as red and blue sticks, respectively. Hydrogen bonds are shown as green dotted lines. Residues forming the canonical PCNA hydrophobic cleft are shown but not labeled. Pol κ PAD and PCNA are shown as ribbons and colored by domain. **c** Pol κ and Pol δ binding to PCNA. The region of Pol κ and Pol δ interacting with PCNA is shown as red and green ribbons, respectively. Interacting PCNA protomer is shown as a light blue surface. **d** Model region of Pol κ active site. Pol κ is shown as a ribbon colored by domains, residues interacting with DNA are shown as sticks. DNA is shown as gray sticks. **e** Side-view of the cryo-EM structures of Pol δ (PDB ID 6TNY)[36] and Pol κ holoenzymes aligned on PCNA. PCNA subunits are shown in different shades of blue and the subunit in the foreground is removed for clarity. DNA molecules in Pol δ and κ structures are shown as yellow and red ribbons, respectively.

the residues responsible for catalysis (Asp107, Asp198 and Glu199) protrude between the triphosphate portion of dTTP and the phosphate group of the terminal templating base (Fig. 2d). The map resolution prevented to discriminate the two metal cations (Ca$^{2+}$), which are normally coordinated in the active site of replicative polymerases. Density at the 5′ end of the template strand is compatible with two bound nucleotides (Fig. 2d). The nucleobase of A at position −1 is packed against the Phe49 in αN2 of the N-clasp, while the nucleobase of T at position −2 is in close proximity to Ser47, which is the last visible residue of the N-clasp. This reinforces the notion that the interaction with DNA is important to stabilize the N-clasp, resulting in the full encirclement of P/T within the Pol κ core[30]. Most of the interactions stabilizing the Pol κ−DNA complex involve the PAD, and are analogous to those reported in the X-ray structure of the Pol κ ternary complex[30].

The B-form dsDNA exiting the Pol κ core bends by ~30°, and threads the PCNA ring with a ~17° tilting angle (Figs. 1f, 2a). The degree of tilting of DNA traversing PCNA is slightly larger than

that observed in the processive Pol δ holoenzyme (~4°), but similar to that observed in the two Pol δ conformers where PCNA is tilted (~16°)[36]. The density of the DNA bases threading the PCNA ring is weak (Fig. 2a), and so is that of the side chains of the basic residues lining the PCNA inner rim, suggesting that the DNA in this region is mobile and the interactions with the DNA phosphates are transient. This observation is consistent with the very low affinity of the PCNA−DNA interaction ($K_d$ ~ 0.7 mM)[37]. The PCNA inner surface, therefore, provides a flexible electrostatic screen for the DNA to pass through unhindered and can adapt to different directions of the duplex DNA leaving the polymerase active site (Fig. 2e). In agreement, recent structures of replicative DNA polymerases of bacteria, archaea and yeast that are bound to both the clamp and DNA show that the DNA traverses the clamp without stable protein−DNA contacts[38−40].

**MD simulations of the Pol κ−PCNA complex in the absence of DNA.** The C-terminal region of the Pol κ PAD is the only

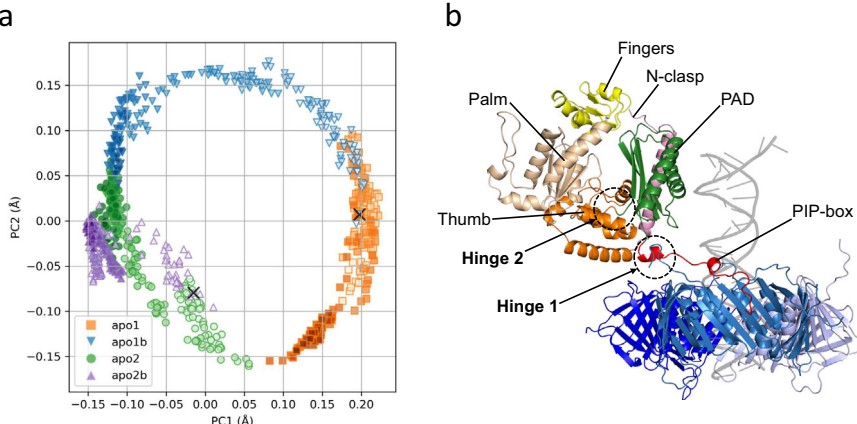

**Fig. 3 MD simulations of the Pol κ−PCNA complex. a** Plot of the projection of the MD trajectories of the Pol κ−PCNA complex onto the first two principal components starting from model apo1 (orange squares and blue triangles) or apo2 (green circles and purple triangles). Intensity of symbol colors is proportional to the time evolution of the trajectories. The cross symbols represent the position at the start of the simulations. **b** Structure of one frame of the apo1 MD trajectory, showing the large changes in the relative orientation of the core and PAD domains of Pol κ and PCNA. Pol κ domains are colored as in Fig. 1, PCNA protomers are colored in different shades of blue. The DNA in the position corresponding to Pol κ holoenzyme is shown as a transparent gray ribbon. The locations of the two hinge regions conferring flexibility to the complex are indicated.

binding site to PCNA and does not participate in any interaction with DNA. The small surface of the PIP-box interaction raises the possibility that, in the absence of DNA, Pol κ bound to PCNA may sample conformations different from that in the holoenzyme. Indeed, flexible binding to the sliding clamp in the apo form was previously suggested for Y-family polymerases Pol IV[41] and Dpo4 [42]. We explored this scenario by performing molecular dynamics (MD) simulations of the Pol κ−PCNA complex based on the cryo-EM structure after removing the DNA (apo1 model) and on the same model but with Pol κ core in the orientation as in the X-ray structure of Pol κ apo form[31] (apo2 model) (Fig. 1b). Two 400-ns simulations for each starting model were performed; a time scale that does not allow equilibrated sampling of the conformational space but can probe flexibility and fast transitioning among potential conformations. Principal component analysis of the MD trajectories shows that the models sample wide yet different conformational space, indicative of large-scale conformational changes (Fig. 3a and Supplementary Fig. 5). Across all simulations, Pol κ maintains the interaction with PCNA via the three PIP-box residues Ile529, Phe532 and Leu533 inserted into the canonical hydrophobic cleft, and Gln526 bound to the Q-pocket. However, Pol κ displays high inter-molecular and inter-domain conformational flexibility due to two flexible hinges, one connecting the PIP-box to the PAD and centered on the inverting helix, and one connecting the PAD to the core domain (Fig. 3b and Supplementary Movies 1−4). While Pol κ overall conformation fluctuates as its domains move around the two hinge regions, Pol κ individual domains and PCNA do not show significant variations, apart from minor shifts in the Pol κ fingers subdomain (Supplementary Movies 1−4). Taken together, the cryo-EM structure and MD simulations suggest that Pol κ bound to PCNA is able to switch from a flexible "carrier state", characterized by high conformational freedom of the core domain relative to the PAD and the PAD relative to PCNA, to a rigid "active state" engaged for DNA synthesis.

**Cryo-EM structure of Pol κ bound to DNA and mono-ubiquitylated PCNA.** In order to reconstitute the Pol κ−DNA−Ub-PCNA complex, we firstly generated PCNA enzymatically mono-ubiquitylated at K164 using the protocol established in Titia Sixma's laboratory, which employs UbcH5c as the E2 ubiquitin-

conjugating enzyme[43]. Purified Ub-PCNA was mixed with Pol κ, the (25/38) P/T DNA containing a dideoxy chain terminator in the primer strand, and dTTP. The complex was separated by micro-SEC, vitrified and subjected to cryo-EM analysis (Supplementary Figs. 6, 7). Refinement of the main 3D class yielded a reconstruction of the Pol κ−DNA−Ub-PCNA complex at 3.7 Å resolution which is nearly identical to that obtained with unmodified PCNA (RMSD$_{Cα}$ ~ 0.5 Å), and where the ubiquitin moieties are not visible (Supplementary Fig. 8). A different 3D class showed residual density protruding radially away from the back face of PCNA where the three lysine residues at position 164 are located (Supplementary Figs. 8, 9). Refinement of this class yielded a map at 6.4 Å resolution, which was further improved by density modification implemented in Phenix[44] (Fig. 4a−c and Supplementary Fig. 7). Additional differences compared to the reconstruction of the complex with wt-PCNA are the appearance of a rod-shaped density corresponding to αN1 of the N-clasp of Pol κ (Fig. 4c and Supplementary Fig. 9), and a slight tilt (~5°) of the catalytic domain towards the PCNA monomer bound to Pol κ (Fig. 4d). Density at the ubiquitin positions is weak and the map local resolution lower than the average (Fig. 4b), implying that the ubiquitins covalently attached to PCNA retain a significant degree of flexibility. However, the density protruding from the PCNA monomer bound to Pol κ is better defined and displays the characteristic barrel-shape of ubiquitin (Ub1, Fig. 4a−c). This may be explained by the fact that the Pol κ-bound monomer of PCNA is more rigid than the other two (Fig. 4b and Supplementary Fig. 7), resulting in a partial restrain of ubiquitin mobility. Rigid-body fitting of ubiquitin into the cryo-EM map at the Ub1 position (Fig. 4c, inset) places ubiquitin in an outward orientation relative to the PCNA ring different from previously published X-ray structures of mono-ubiquitylated PCNA[45,46] (Fig. 4e). However, due to the weak map density and local resolution in the Ub1 region, we cannot exclude that ubiquitin may exist in multiple orientations which the cryo-EM reconstruction could not resolve. The outward location of ubiquitin density implies the exposure of the ubiquitin hydrophobic patch to interact with Pol κ UBZs, but the map features are at odds with the existence of a rigid Ub/UBZ complex (Fig. 4a−c), suggesting that the Ub/UBZ interaction is transient and/or involves multiple ubiquitin and UBZ conformations.

Collectively, these data show that (i) the position of Pol κ relative to Ub-PCNA in the complex is entirely dictated by the

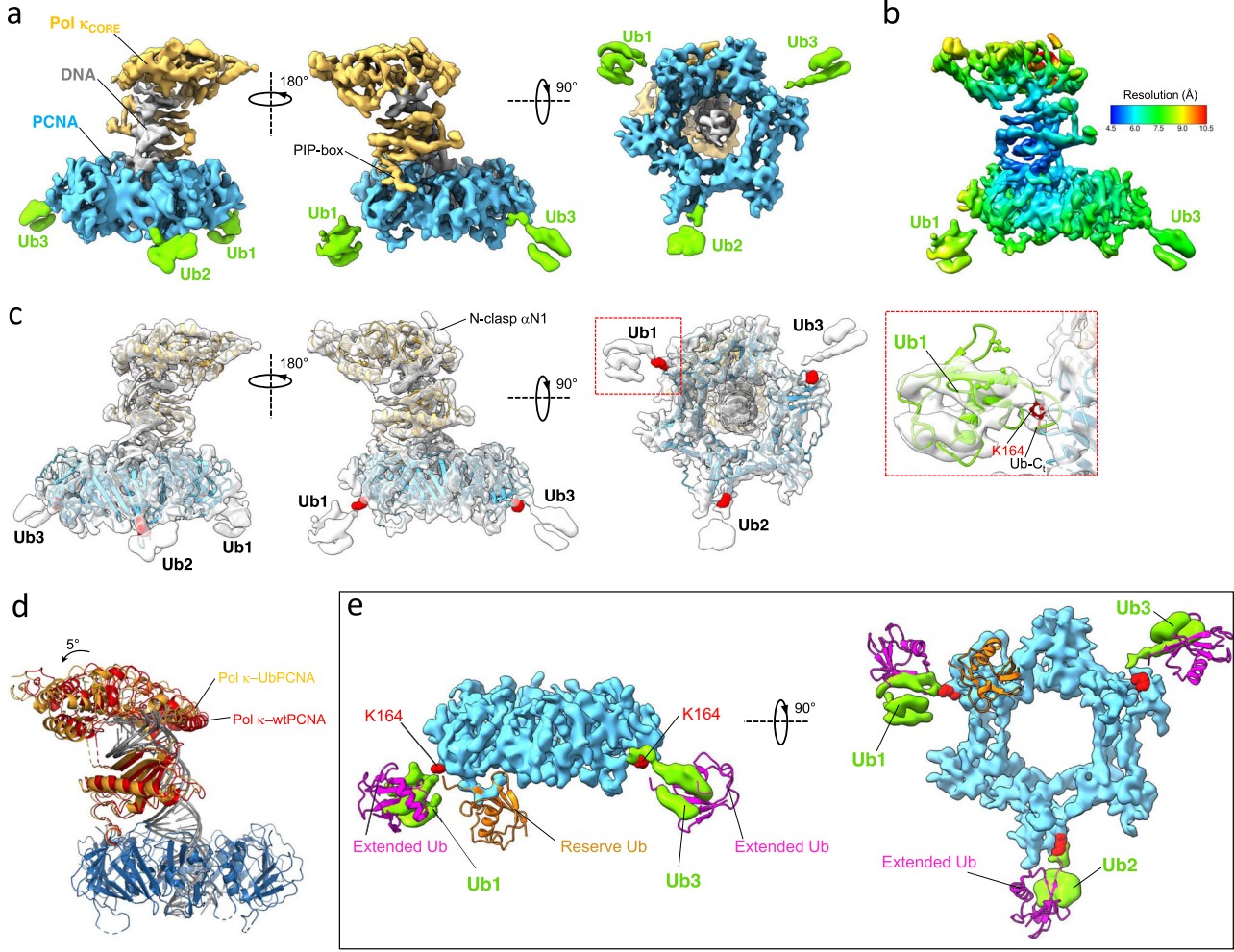

**Fig. 4 Cryo-EM structure of the Pol κ−DNA−Ub-PCNA complex. a** Different views of the 6.4 Å reconstruction colored by components (Pol κ in orange, PCNA in skyblue, DNA in gray and ubiquitin in green). **b** Cryo-EM map color-coded by local resolution. **c** Structural model fitted into the cryo-EM map. The lysine residues at position 164 are shown as red spheres. The inset shows the rigid-body fitting of ubiquitin structure into the cryo-EM map at Ub1 position. The ubiquitin L8/I44/V70 residues at the hydrophobic patch, and K164 on PCNA are shown as sticks. **d** Overlay of structural models of Pol κ bound to wt-PCNA or Ub-PCNA aligned on PCNA, highlighting the slight difference in Pol κ core tilt. **e** Fitting of crystal structures of yeast split Ub-PCNA with ubiquitin in the "reserve" position (PDB ID 3L0W)[45], and human Ub-PCNA with ubiquitin in the "extended" position (PDB ID 3TBL)[46] into the cryo-EM map of the Pol κ−DNA−Ub-PCNA complex. Cryo-EM densities of PCNA and ubiquitin are colored in skyblue and green, respectively. Map portions of Pol κ and DNA were removed for clarity. The lysine residues at PCNA position 164 are shown as red spheres.

internal PIP-box interaction and by the interaction of Pol κ with DNA, (ii) the ubiquitins conjugated to PCNA are flexible, with their hydrophobic patch mostly accessible to interact with Pol κ UBZs, and (iii) the C-terminal region of Pol κ encompassing the two UBZs is flexible and is not rigidified upon binding to Ub-PCNA, implying that the interaction between the UBZs and ubiquitin is transient and/or comprises various orientations of the ubiquitin moieties.

**DNA replication assays with Pol κ mutants.** In order to functionally validate the structural features observed in the cryo-EM reconstructions, we carried out primer extension assays by Pol κ−PCNA using Pol κ variants containing mutations at the PCNA interface and in the two UBZs (Fig. 5a). Six Pol κ mutants were expressed and purified to homogeneity (Supplementary Fig. 10) and their activity was tested on a primer/template DNA substrate with blocked ends and on which wt-PCNA or Ub-PCNA had been previously loaded by the RFC clamp loader (Fig. 5b). The quantitation of Pol κ mutants' synthetic activity relative to wild-

type Pol κ in this assay is described in the "Methods" section and reported in Supplementary Table 2. We found that wild-type Pol κ was similarly stimulated by wt-PCNA and Ub-PCNA in synthesizing the full primer (Fig. 5c, lanes 3 and 4, Fig. 5e and Supplementary Table 2). Two helix-breaking proline mutations at the center of the Pol κ inverting helix (DR mutant) only slightly affected Pol κ synthetic activity (Fig. 5c, lanes 12 and 13, Fig. 5e and Supplementary Table 2), suggesting that alternate conformations in this region may function equally well to poise the internal PIP-box of Pol κ to interact productively with PCNA. Conversely, mutation of key residues in the internal PIP-box (IFL or QRSI mutants) critically impaired Pol κ activity with wt-PCNA (Fig. 5c, lanes 6 and 9, Fig. 5e and Supplementary Table 2), confirming the structural observation that the internal PIP-box is the key determinant for the Pol κ−PCNA interaction. Notably, the PIP-box mutants only partially decreased Pol κ activity when Ub-PCNA was present (Fig. 5c, lanes 7 and 10, Fig. 5e and Supplementary Table 2), suggesting that, when the internal PIP-box interaction is disrupted, a secondary interaction with ubiquitin contributes in retaining Pol κ on PCNA. To confirm this

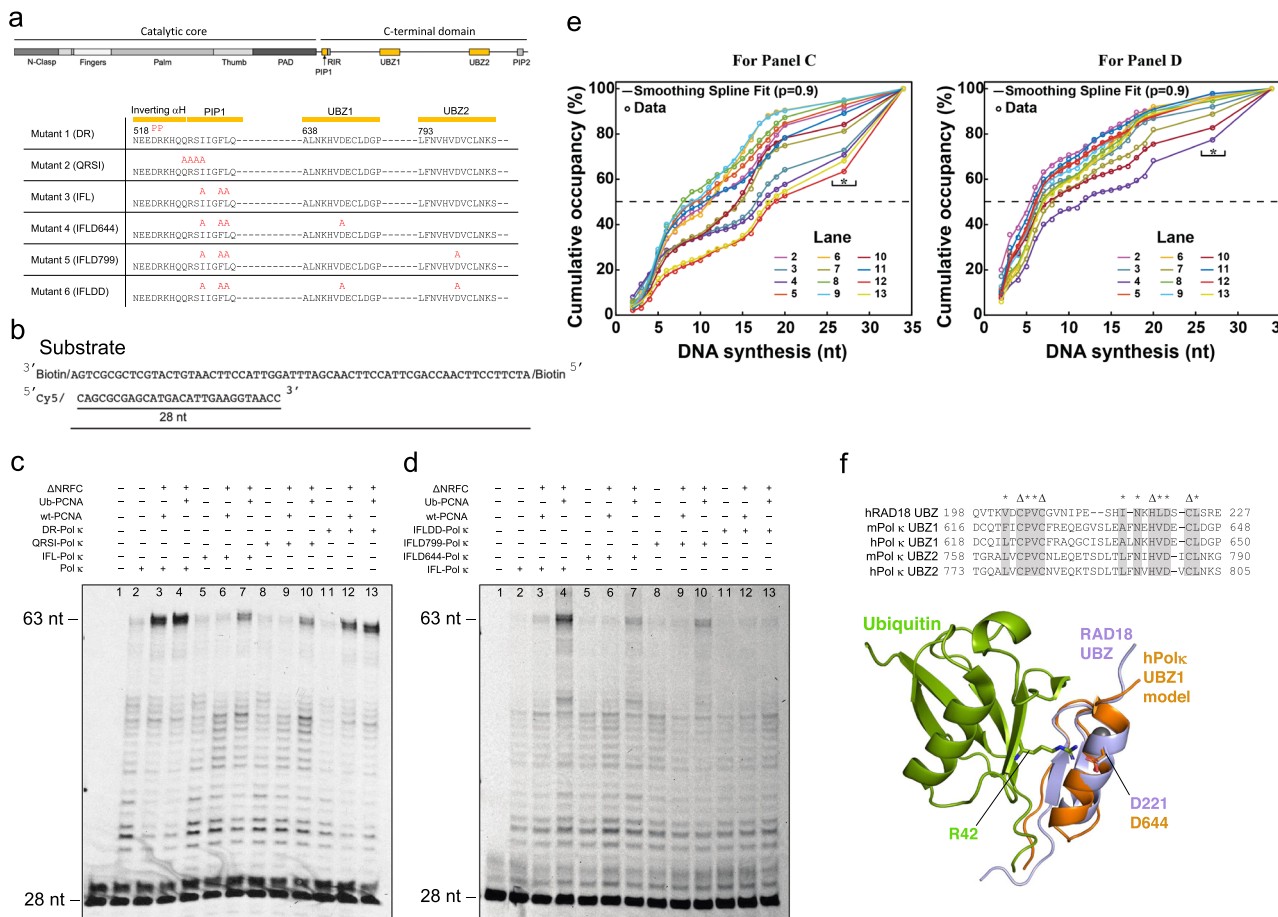

**Fig. 5 Mutational analysis of the Pol κ synthetic activity. a** Schematic view of the Pol κ mutants used in the primer extension assays. Mutated residues are colored red above the native sequence. **b** Schematic representation of the primer/template DNA substrate used in the assays. **c**, **d** Activity of Pol κ and stimulation by PCNA on the DNA substrate. Wild-type Pol κ and Pol κ mutants were incubated with the substrate in the presence and absence of PCNA at 30 °C for 40 s and the products were separated on 15% denaturing polyacrylamide gels at 12 W for 3 h. The reaction's procedure is detailed in the "Methods" section. Experiments have been repeated at least two times. **e** Plots of the cumulative percentages of DNA synthesis up to the indicated number of synthesized nucleotides constructed from the lanes of the gels presented in panels (**c**) and (**d**), respectively. The curves were generated as described in the "Methods" section. The experimental data points (circles) were fitted to relatively rough smoothing splines (solid lines) with the indicated smoothing parameter. The curves account exclusively for the distribution of the bands corresponding to DNA synthesis products and not for the remaining enzymatically unmodified substrate. In both graphs, the horizontal black dashed lines represent a median DNA synthesis value of 50%. The intersection of each smoothing spline fit with this 50% line gives an $N_{1/2}$ value, which represents an apparent number of synthesized nucleotides up to which 50% of the total DNA synthesis is achieved. For both graphs, the experimental data points and their corresponding fitting curves are color-coded as presented in the inset legend. In both graphs, the black star sign marks the values corresponding to 27 synthesized DNA nt to highlight that this value was not obtained directly but rather represents a summation of the smeared region between 21 and 33 nt following background subtraction. **f** (top) Alignment of the sequences of human Rad18-UBZ domains and Pol κ UBZ domains from mouse and human. Zinc-coordinating residues are labeled with Δ. Residues at the interface with ubiquitin are labeled with asterisks. Conserved residues by type are colored gray. (bottom) Homology model of human Pol κ UBZ1 aligned on the NMR structure of Rad18-UBZ/Ubiquitin complex (PDB ID 2MRE)[51]. Pol κ UBZ1 model was built with HHpred[95] based on the Rad18-UBZ/ubiquitin complex structure[51]. Residues involved in the salt bridge critical for UBZ/Ubiquitin complex formation are shown as sticks.

possibility, we designed three Pol κ mutants containing mutations in both the internal PIP-box and in either one or both Pol κ UBZs (Fig. 5a). The UBZ mutation (D644A in UBZ1 and D799A in UBZ2) was selected based on the sequence conservation between the mouse and human Pol κ UBZ orthologs as well as the UBZ of human RAD18 (Fig. 5f). NMR studies showed that RAD18 UBZ binds ubiquitin with micromolar affinity through the ubiquitin hydrophobic patch centered on L8/I44/V70, and that the conserved aspartic acid C-terminal to the zinc-coordinating histidine establishes a salt bridge with ubiquitin Arg42, which is critical for complex formation[44] (Fig. 5f). Consistently, previous biochemical data in mouse showed that alanine mutation of the conserved aspartate in both UBZs disrupted the association between Pol κ and ubiquitin, the association between Pol κ and Ub-PCNA, as

well as the association of Pol κ with replication factories in cells exposed to UV radiation[45]. In our assay, the synthetic activity of Pol κ variants bearing mutations in the PIP-box and one of the two UBZs (ILFD644 and ILFD799 mutants) with Ub-PCNA (Fig. 5d, lanes 7 and 10, Fig. 5e and Supplementary Table 2) was reduced compared to the PIP-box mutant only (Fig. 5d, lane 4, Fig. 5e and Supplementary Table 2), while mutation of the PIP-box and both UBZs (Mutant IFLDD) resulted in minimal synthetic activity (Fig. 5d, lane 13, Fig. 5e and Supplementary Table 2). The effects of UBZ mutations are not observed when PCNA is not ubiquitylated, demonstrating that the interaction mediated by the UBZs is ubiquitin-specific. To confirm a direct interaction between Pol κ UBZ1/2 and ubiquitin, we used NMR and measured the chemical shift perturbations of [15]N-labeled

ubiquitin in the presence of a ten-fold excess of unlabeled synthetic peptides encoding either UBZ1 or UBZ2 (Supplementary Figs. 11, 12). Chemical shift mapping showed that both UBZ1 and UBZ2 domains interact with the conserved hydrophobic surface of the ubiquitin β-sheet centered at L8/I44/V70 (Supplementary Fig. 12), as observed for Rad18 UBZ[44]. Perturbations are larger for UBZ1 (Supplementary Fig. 12), suggesting a higher binding affinity compared to UBZ2. Additionally, many of the ubiquitin resonances in the presence of UBZ1 become broader and weaker indicating they are approaching the intermediate exchange regime. This broadening was not observed at the equivalent concentration of UBZ2, pointing to a lower occupancy of the binding site which is associated with a lower affinity interaction. However, due to the physical properties of both peptides and limiting solubility, it is not possible to accurately determine the concentration of either peptide, so making an accurate estimate of binding affinities impossible.

Collectively, our data strongly suggest that the PCNA-enhanced activity of Pol κ is controlled primarily by the internal PIP-box, which overrides the secondary interaction between the UBZs and ubiquitin. When the PIP-box interaction is lost, the UBZ/ubiquitin interaction becomes significant in retaining Pol κ on PCNA and preventing its detachment from the DNA template.

## Discussion

### Functional implications of the Pol κ−DNA−PCNA complex structures.

The structure of the Pol κ−DNA−wt-PCNA complex provides a near-atomic resolution structure of a Y-family DNA polymerase bound to its processivity factor and DNA. The most striking feature consists in the sharply angled orientation of the Pol κ core relative to the PCNA ring, and the resulting bending of dsDNA threading the PCNA central hole. The interaction tethering Pol κ to PCNA in this tilted position is mediated by the internal PIP-box ($^{526}$QRSIIGFL) adjacent to the PAD of Pol κ, while the PIP-box ($^{862}$KHTLDIFF) at the extreme C-terminus does not participate in the interaction. This is in agreement with our functional analysis showing that mutation of key residues in the internal PIP-box impairs Pol κ−PCNA processivity, and is consistent with previous studies reporting that the terminal PIP-box does not modulate Pol κ synthetic activity[19]. While a fragment encoding the terminal PIP-box of Pol κ has been co-crystallized with PCNA[47], binding of this fragment to PCNA in a surface plasmon resonance assay was not observed[47], suggesting that the interaction is extremely weak.

Y-family polymerases Pol η, ι, and κ display a high degree of domain conservation, and all three possess internal PIP-boxes adjacent to the PAD[13]. Therefore, the structural features observed in the Pol κ−DNA−PCNA complex are likely general and may apply to the corresponding complexes with Pol η and ι. Indeed, the PCNA-stimulated processivity of both Pol η and ι is dependent on internal PIP-boxes located at the C-terminus of the PAD[19]. As we propose later, the tilted position of Pol κ may be required to accommodate the replicative polymerase on PCNA during polymerase exchange in TLS.

The internal PIP-box is the only binding site of Pol κ to PCNA, involves a relatively small interaction surface, and does not contact DNA. In the absence of DNA, our MD simulations predict that Pol κ bound to PCNA samples a wide conformational space, due to the conformational freedom of the core domain and PAD, which results in an ensemble of different orientations of Pol κ relative to PCNA (carrier state). Binding to P/T DNA locks the polymerase−clamp complex into a rigid conformation that is competent for catalysis (active state). Thus, PCNA may facilitate the recruitment of the "carrier state" polymerase to damaged P/T

junctions when the polymerase is located away from the target P/T junction. Binding of the Pol κ PAD to PCNA encircling dsDNA, and the disengagement of the Pol κ core from duplex DNA, would ensure a rapid relocation of the polymerase to the target P/T junction due to the fast 1D diffusion of PCNA on dsDNA (diffusion coefficient ~ 1 $\mu m^2 \, s^{-1}$)[48,49].

The long Pol κ region C-terminal to the internal PIP-box (residues 535−870) is invisible in the cryo-EM map, suggesting it is largely disordered, in agreement with disorder predictions[29] (Fig. 1a). In Pol η, the flexibility of the C-terminal region has been previously observed experimentally[50], and appears as a common characteristic of eukaryotic Y-family polymerases[29]. The disordered C-terminal regions of Pol η and Pol κ contain one and two UBZs, respectively, while that of Pol ι bears two ubiquitin binding motifs (UBMs). Both UBZs of Pol κ display a striking sequence similarity with Rad18-UBZ[51], which binds ubiquitin with micromolar affinity through the canonical hydrophobic patch centered on L8/I44/V70[51]. In a current functional model, the UBZs or UBMs of TLS polymerases bind the ubiquitin moieties covalently attached to PCNA mono-ubiquitylated at K164 by Rad6–Rad18[45], aiding their recruitment to sites of DNA damage and the replacement of the stalled high-fidelity polymerase[52–54]. However, the structural basis of the interaction of TLS polymerases with mono-ubiquitylated PCNA has so far remained elusive. The structure of mono-ubiquitylated PCNA in isolation has been previously investigated in several reports. These include a crystal structure of a yeast PCNA split molecule where the N-terminus of PCNA was co-expressed with a linear fusion of ubiquitin with the C-terminus of PCNA[45], a crystal structure of human PCNA enzymatically ubiquitylated in vitro with RNF8 and UbcH5[46], a study combining small-angle X-ray scattering (SAXS) and molecular modeling[55], and a study combining SAXS and NMR[43]. The picture emerging from this work is that the ubiquitins covalently bound to PCNA are flexible and can dynamically switch among an ensemble of different orientations. Discrete positions such as that observed in the split PCNA structure where ubiquitin is bound to the back of PCNA[45] were suggested to pertain to a "reserve state" in which the TLS polymerase is held next to PCNA but is disengaged from the P/T DNA junction[55]. Alternate positions in which ubiquitin extends away from the PCNA ring were instead suggested to exist in the DNA-bound, active state of the TLS polymerase[55]. However, since these studies were performed in the absence of a TLS polymerase and DNA, a functional assignment of the different ubiquitin conformations remained ambiguous. Based on a 22-Å-resolution EM structure of Ub-PCNA bound to Pol η and DNA, it has been proposed that Pol η binding to Ub-PCNA results in the formation of a structured interface between Pol η C-terminal region and one of the three ubiquitins attached to Ub-PCNA[56]. Due to the map's low resolution, however, the location of both the polymerase and ubiquitin moieties relative to PCNA remained underdefined[56]. Our cryo-EM structure of PCNA enzymatically mono-ubiquitylated at K164 in complex with both Pol κ and DNA shows that the orientation of Pol κ relative to PCNA is dictated by the PIP-box and DNA interactions, and that the three ubiquitins are partly flexible and extend away from the PCNA ring. The position(s) of ubiquitin attached to the PCNA monomer bound to Pol κ appears different from those in the reported X-ray structures of Ub-PCNA[45,46], suggesting that crystallization of the flexible ubiquitin may have selected specific conformations that are favored by crystal contacts. While in the cryo-EM reconstruction ubiquitins are expected to have their hydrophobic patch accessible to interact with Pol κ UBZs, the formation of a rigid ubiquitin/UBZ complex is not observed, suggesting that the interaction with ubiquitin is dynamic and does not generate a structured interface.

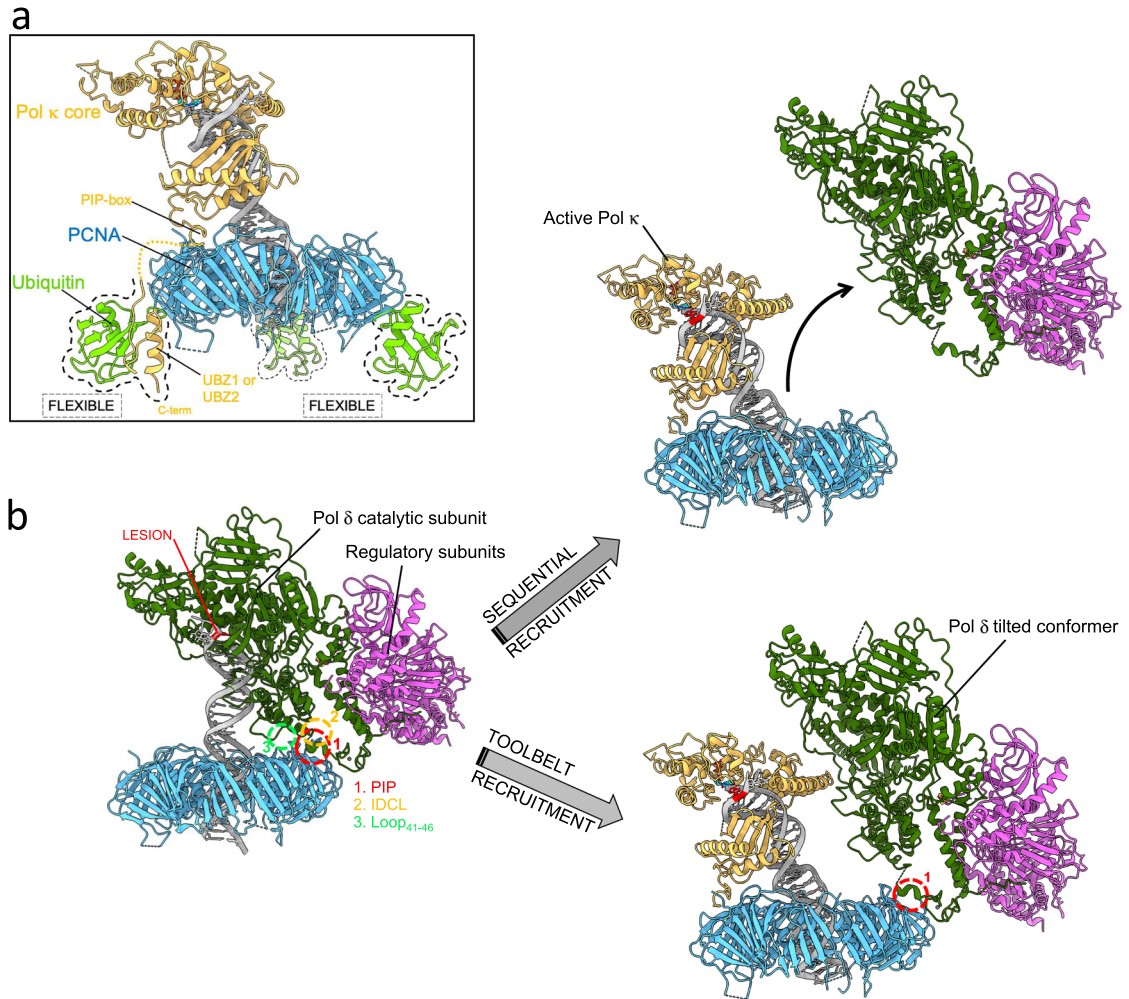

**Fig. 6 Proposed models of function of Pol κ in TLS. a** Pol κ interaction with mono-ubiquitylated PCNA. Pol κ binds PCNA encircling DNA through the internal PIP-box at the C-terminus of the PAD. If the PIP-box interaction is lost, Pol κ UBZ zinc fingers interact with the ubiquitin molecules flexibly attached to the back face of Ub-PCNA, helping retain the polymerase to the DNA primer/template junction. Ubiquitins are modeled as in the cryo-EM structure of Pol κ—DNA—Ub-PCNA complex (Ub1 position), but may occupy alternate positions due to their intrinsic flexibility. The orange dotted line represents the C-terminal disordered region connecting the Pol κ core to the UBZ domains. The homology model for Pol κ UBZ/Ubiquitin complex was built as shown in Fig. 5. **b** PCNA-directed polymerase swapping in TLS. At a lesion on the DNA template strand (highlighted in red), Pol δ holoenzyme stalls, and Pol κ is recruited to PCNA following two alternative paths. Either Pol δ dissociates into solution prior to or during Pol κ binding to PCNA and P/T DNA (sequential model), or Pol δ tilts remaining attached to the PIP-box site on PCNA and Pol κ is recruited to the exposed PIP-box site, capturing the P/T DNA released by Pol δ (toolbelt model). Tilting of Pol δ is achieved by disruption of two of the three indicated contact points with PCNA[36], and allows to accommodate actively synthesizing Pol κ on PCNA in the "active state" without steric clashes.

Previous biochemical work showed that the UBZs of the mouse ortholog of Pol κ are (i) required for the association between Pol κ and ubiquitin, (ii) important for the association between Pol κ and Ub-PCNA, and (iii) required for association of Pol κ with replication factories in cells exposed to UV radiation[53]. In agreement with these findings, our primer extension data show that mutation of a conserved aspartate in either of the two UBZs, which is expected to impair the binding to ubiquitin[51,53], significantly decreases the processivity of human Pol κ when PCNA is mono-ubiquitylated and the primary PIP-box interaction is disrupted. Thus, the PCNA-dependent activity of Pol κ is primarily controlled by the internal PIP-box interaction and, in case such interaction is lost, by a secondary interaction between the UBZs and the ubiquitin flexibly attached to Ub-PCNA. The existence of this Ub/UBZ secondary interaction is fully compatible with our structural model, since the long flexible C-terminus of Pol κ may easily bring the UBZs in proximity to one or more ubiquitins attached to the PCNA homotrimer (Fig. 6a).

In the human system, the Ub/UBZ secondary interaction has been recently challenged by in vitro experiments using primer extension assays by the replicative polymerase Pol δ showing that the TLS activity of Pol η and κ is independent of PCNA ubiquitylation[57–59]. In vivo, however, the interaction mediated by ubiquitin may help Pol η or Pol κ to outcompete other proteins present at the replication fork, such as FEN1, Lig1 and PAF15, which all bind PCNA via similar PIP-box interactions[60–62]. In fact, conflicting observations on the role of PCNA ubiquitylation were also made in yeast using primer extension assays by Pol δ[63,64]. Nonetheless, a recent study showed that PCNA ubiquitylation stimulates the recruitment of Pol η in a fully reconstituted yeast replisome, underlining the importance of studying the role of PCNA ubiquitylation in TLS within the context of the replisome[26].

**Implications for polymerase swapping in TLS.** Eukaryotic Pol δ bound to PCNA replicates the DNA lagging strand, and is also critical for recoupling of leading-strand synthesis to the CMG helicase following lesion bypass[26], but its synthetic activity and fidelity are impaired by damaged templates, particularly containing bulky lesions or abasic sites[65]. How a damaged DNA template may be transferred from stalled Pol δ to a TLS polymerase to restart synthesis is unclear. Biochemical experiments showed that the higher affinity of Pol η for P/T DNA relative to Pol δ drives the exchange of the two polymerases in human TLS, and that PCNA is retained on the DNA substrate during the competition[57]. This agrees with a recent live-cell imaging analysis showing that, for the length of time PCNA is retained on damaged DNA, Pol δ and a TLS polymerase could have exchanged around 60 times[66].

Considering the structures of actively synthesizing Pol δ−PCNA[36] and Pol κ−PCNA (current study), it follows that Pol δ needs to move away from the P/T junction to grant access to Pol κ. Two scenarios are possible for polymerase exchange: either stalled Pol δ completely dissociates from PCNA and Pol κ is recruited (sequential model), or Pol δ and Pol κ co-exist on the same PCNA ring during the DNA handoff (toolbelt model) (Fig. 6b). The second possibility would require tilting of Pol δ after the release of DNA. We have previously described tilting of Pol δ relative to PCNA in two Pol δ holoenzyme alternate cryo-EM conformers[36]. It is, therefore, possible that Pol δ may lose the interactions with the IDCL and Loop$_{41-46}$ of PCNA and retain the PIP-box interaction, resulting in a PCNA tilting which provides enough room to accommodate actively synthesizing Pol κ without steric clash (Fig. 6b). In fact, Pol δ in the tilted conformers is bound to PCNA only via the PIP-box[36], suggesting that in the absence of DNA, it will be flexibly tethered to PCNA and sampling different conformations. Interestingly, biochemical experiments showed the coexistence of Pol III and Pol IV on β-clamp in bacterial TLS[67,68], and a supra-holoenzyme consisting of PolB1 and PolY simultaneously bound to PCNA has been characterized in archeal TLS[69]. If a "TLS toolbelt" involving Pol δ exists in eukaryotes remains to be determined. We attempted to reconstitute a Pol δ−Pol κ−PCNA−DNA toolbelt and image it by cryo-EM but failed to observe both Pol δ and Pol κ bound to a single PCNA ring (Supplementary Fig. 13). However, it is possible that such a toolbelt is short-lived and could not be resolved in these experiments. In addition, ubiquitylation of PCNA may be required to retain Pol κ on PCNA during polymerase swapping. Our previously reported structure of human Pol δ and FEN1 simultaneously bound to PCNA[36] provides direct evidence of a toolbelt in eukaryotes. It is, therefore, possible that a TLS polymerase may replace FEN1 on PCNA to perform lesion bypass. Ubiquitylation of PCNA may help recruit the TLS polymerase allowing it to outcompete FEN1. Further structural and functional studies on a fully reconstituted eukaryotic lagging-strand replisome are needed to explore these possibilities.

The functional role of the sharply angled orientation of Pol κ core relative to the PCNA ring and the resulting bending of dsDNA is unclear. It is possible that DNA bending might be important to sterically clear the front face of PCNA to retain Pol δ in a toolbelt mechanism or recruit it from solution. It might also be possible that it acts to strike a balance between the need for PCNA to recruit and stimulate the activity of Pol κ and the release of Pol κ after it bypasses the lesion. In this latter scenario, the bent DNA is not the optimal conformation for nucleotide incorporation by Pol κ, which may counterbalance the enhanced affinity of Pol κ to DNA when bound to PCNA. The flexibility in the orientation of the DNA in the PCNA ring in Pol κ−PCNA and Pol δ−PCNA structures[36] suggests that the inner surface of PCNA can provide a flexible electrostatic screen for the DNA to pass through unhindered and support tilting conformers of the exiting DNA if necessary.

## Methods

**Protein expression and purification.** Human Pol κ was purified using a modified version of previously published protocol[70]. *E. coli* codon-optimized sequence of full-length human Pol κ (accession no. NP057302) was cloned into a pE-SUMOpro expression vector (Lifesensors) using Gibson assembly technology. Different Pol κ mutants were generated by PCR separately. PIP-box domain mutants: Q526, R527, S528 and I529 to AAAA and I529, F532 and L533 to AAA named hereafter as QRSI-Pol κ and IFL-Pol κ, respectively. UBZs domain mutants in IFL-Pol κ background include D644A, D799A, D644, D799 to AA double mutant named hereafter IFLD644-Pol κ, IFLD799-Pol κ and IFLDD-Pol κ, respectively. D521P and R522P double mutant was also generated by PCR named hereafter DR-Pol κ.

The Pol κ and different mutant plasmids were transformed into *E. coli* strain BL21 (DE3) competent cells (Novagen) that were grown on agar plates containing 50 μg/ml kanamycin. Several colonies were randomly selected and checked for expression level. Pol κ was overexpressed by growing the transformed cells into 10 l of 2YT media (Teknova) supplemented with kanamycin. Cells were incubated at 24 °C till the $OD_{600}$ reached 0.8 and then protein expression was induced at 0.1 mM isopropyl β-D-thiogalactopyranoside (IPTG) concentration. The cells were incubated further for 19 h at 16 °C, harvested by centrifugation at $5500 \times g$ for 10 min, then re-suspended in lysis buffer [50 mM Tris pH (8), 750 mM NaCl, 40 mM imidazole, 5 mM β-mercaptoethanol, 0.2% NP-40, 1 mM PMSF, 5% glycerol and EDTA free protease inhibitor cocktail tablet/50 ml (Roche, UK)]. All subsequent steps were performed at 4 °C. The cells were lysed by 1 h incubation on ice using lysozyme at a final concentration of 2 mg/ml followed by mechanical disruption by sonication. Cell debris was then removed by centrifugation at $22,040 \times g$ for 30 min. The decanted supernatant was directly loaded onto HisTrap HP 5 ml affinity column (GE Healthcare) pre-equilibrated with Buffer A [50 mM Tris (pH 7.5), 500 mM NaCl, 40 mM imidazole, 5 mM β-mercaptoethanol and 5% glycerol]. The column was washed with ten column volumes (CVs) of Buffer A and eluted by ten CV gradient against Buffer B [50 mM Tris (pH 7.5), 500 mM NaCl, 500 mM imidazole, 5 mM β-mercaptoethanol and 5% glycerol]. The protein was eluted around 210 mM imidazole concentration. The fractions containing Pol κ were checked by SDS-PAGE. The peak fractions were then pooled and SUMO protease was added to cleave the SUMO tag and generate Pol κ in the native form and dialyzed overnight against dialysis buffer [50 mM Tris (pH 7.5), 500 mM NaCl, 5 mM β-mercaptoethanol and 5% glycerol]. The dialyzed sample was then loaded again onto HisTrap HP 5 ml column using Buffer A and the native protein was collected in the flow-through fractions. Fractions that contained Pol κ or mutants were pooled, concentrated and then loaded onto HiLoad 16/600 Superdex 200 pg (GE Healthcare) equilibrated with storage buffer [50 mM Tris (pH 7.5), 300 mM NaCl and 1 mM DTT]. Fractions containing Pol κ or mutants were checked for purity, concentrated, flash frozen and stored at −80 °C.

Human PCNA used for the Pol κ and its mutants in replication assays was produced as described previously[36]. Briefly, full-length human PCNA (accession no. NM182649) was cloned into pETDuet-1 MCS1 (Novagen) Amp+ to obtain 6× His N-terminally tagged protein, transformed into *E. coli* strain BL21 (DE3) cells and grown at 37 °C in 2YT media supplemented with ampicillin till reaching $OD_{600}$ of 1.2. Protein expression was induced with 0.5 mM IPTG for 19 h at 16 °C. Cells were then harvested, and lysed with lysozyme followed by sonication. The cleared lysate was loaded onto a HisTrap column (GE Healthcare) and eluted with low salt, followed by anion exchange on a HiTrap Q column (GE Healthcare), and finally size exclusion chromatography on a HiLoad 16/600 Superdex 200 pg. Pure protein fractions were pooled, flash-frozen, and stored at −80 °C. Recombinant PCNA used for the Cryo-EM study was produced as described previously[37].

Ub-PCNA was produced and purified as described previously[43] with slight modifications. For Ub-PCNA purification, human ubiquitin, PCNA, E1 (Uba1) and UbcH5c (S22R) were expressed in BL21 (DE3) *E. coli* cells. Human ubiquitin, E1 (Uba1) and UbcH5c (S22R) were expressed by growing the transformed cells into LB medium supplemented with respective antibiotics at 37 °C. When cell densities reached an absorbance of 0.6 at 600 nm, they were induced with a final concentration of 0.5 mM IPTG and left to express at 18 °C overnight. Cells were then harvested, lysed and purified by successive chromatography steps. Ubiquitin expressed cells were suspended in lysis buffer [50 mM HEPES pH 7.5, 150 mM NaCl, 1 mM DTT, DNAse and EDTA free protease inhibitor cocktail tablet (Roche, UK)]. The cells were lysed by sonication and cell debris was removed by centrifugation at $18,000 \times g$ for 45 min. The supernatant was incubated at 90 °C for 10 min followed by centrifugation at $13,000 \times g$ for 30 min. The supernatant was pooled, concentrated and loaded onto Superdex 75 10/300 GL (GE Healthcare) equilibrated with storage buffer [20 mM Tris pH 8, 150 mM NaCl, 0.5 mM TCEP]. Fractions containing ubiquitin were pooled, concentrated, flash frozen and stored at −80 °C.

E1 (Uba1) expressed cells were suspended in lysis buffer [50 mM HEPES pH 7.5, 150 mM NaCl, 0.5 mM TCEP, DNAse and EDTA free protease inhibitor cocktail tablet (Roche, UK)]. The cells were lysed by sonication and cell debris was removed by centrifugation at $18,000 \times g$ for 45 min. The decanted supernatant was directly loaded onto HisTrap HP 1 ml column (GE Healthcare) pre-equilibrated with Buffer A [50 mM HEPES pH 7.5, 150 mM NaCl, 0.5 mM TCEP, 10 mM

imidazole]. The column was washed with ten CVs of Buffer A and eluted by five CV of Buffer B [50 mM HEPES pH 7.5, 150 mM NaCl, 1 mM TCEP, 250 mM imidazole]. The fractions containing E1 were checked by SDS-PAGE, pooled and dialyzed overnight against dialysis buffer [20 mM HEPES, pH 7.5, 150 mM NaCl, 1 mM TCEP]. The dialyzed sample was loaded again onto HiTrap Q HP 5 ml column pre-equilibrated with the same buffer and eluted by 20 CV gradient against Buffer B [20 mM HEPES 7.5, 1 M NaCl, 1 mM TCEP]. The E1 was eluted 50% of Buffer B. The fractions containing E1 were pooled, concentrated and loaded onto Superdex 200 10/300 GL (GE Healthcare) equilibrated with storage buffer [20 mM Tris pH 8, 150 mM NaCl, 0.5 mM TCEP]. Fractions containing E1 were pooled, concentrated, flash frozen and stored at −80 °C.

H5C expressed cells were suspended in lysis buffer [50 mM HEPES pH 7.5, 150 mM NaCl, 0.5 mM TCEP, 10 mM imidazole, lysozyme, DNAse and EDTA free protease inhibitor cocktail tablet (Roche, UK)]. The cells were lysed by sonication and cell debris was removed by centrifugation at $18,000 \times g$ for 45 min. The decanted supernatant was directly loaded onto HisTrap HP 1 ml affinity column (GE Healthcare) pre-equilibrated with Buffer A [50 mM HEPES pH 7.5, 150 mM NaCl, 0.5 mM TCEP, 10 mM imidazole]. The column was then washed with ten CVs of Buffer A and eluted by five CV of Buffer B [50 mM HEPES pH 7.5, 150 mM NaCl, 1 mM TCEP, 250 mM imidazole]. The fractions containing H5C were checked by SDS-PAGE, pooled and dialyzed overnight against dialysis buffer [50 mM HEPES pH 7.5, 150 mM NaCl, 1 mM TCEP]. The dialyzed sample was then concentrated and loaded onto Superdex 75 10/300 GL (GE Healthcare) equilibrated with storage buffer [50 mM HEPES pH 7.5, 150 mM NaCl, 1 mM TCEP]. Fractions containing H5C were pooled, concentrated, flash frozen and stored at −80 °C.

Ub-PCNA reaction was set up using 80 nM E1, 32 µM UbcH5c (S22R), 32 µM ubiquitin, and 20 µM PCNA in a buffer containing 50 mM malic acid-MES-Tris (MMT) buffer, pH 9.0, 25 mM NaCl, 3 mM MgCl$_2$, 0.5 mM TCEP, and 3 mM ATP. Experiments were performed for 2 h at 37 °C in a 10 µl volume initially and confirmed the Ub-PCNA formation by SDS-PAGE. For Ub-PCNA purification, the ubiquitylation reaction was finally set up for 1 ml and the product was finally purified using a Superdex 200 10/300 GL (GE Healthcare) size exclusion chromatography column using 10 mM HEPES, pH 7.5, 150 mM NaCl, 0.5 mM TCEP as the running buffer. Fractions containing Ub-PCNA were pooled, concentrated, flash frozen and stored at −80 °C. The incorporation of ubiquitin to PCNA was confirmed by LC-MS using Xevo G2-XS QTof mass spectrometer (Waters) coupled to an Acquity LC system (Waters) using an Acquity UPLC Protein BEH C4 column (2.1 × 100 mm, Waters).

Ubiquitin for the NMR experiments was cloned into an expression vector pET28a and the recombinant protein was overexpressed in *E. coli* strain BL21 (DE3). Uniformly $^{15}$N samples were grown in modified Spizizen's media with 1.0 g/l of $^{15}$N ammonium chloride as the sole nitrogen source. Bacterial cultures were grown at 37 °C to an optical density of ~0.7, whereupon the temperature of the culture was reduced to 20 °C and protein expression was induced by the addition of IPTG to a final concentration of 0.450 mM. Cultures were grown for a further 16 h. Cells were harvested and then lysed into 50 mM Tris, pH 8.0. Initial purification (~70%) was achieved by affinity chromatography using Ni–NTA resin. After TEV cleavage of the affinity tag and dialysis into 20 mM phosphate buffer pH 6.5, 100 mM NaCl, the protein was further purified by size exclusion chromatography using a Superdex S75 column (GE Life Sciences). The protein was concentrated and its purity assessed by Coomassie-stained SDS-PAGE.

Human Pol κ UBZ1 (residues 618−650) and UBZ2 (residues 773−805) synthetic peptides (>98%purity) termini-blocked by N-acetyl and C-amide groups were purchased from Thermo Fisher.

**DNA substrates.** DNA oligos for the primer extension assays were synthesized and HPLC purified by IDT and The Midland Co. The non-damage substrate consisted of a 63 nt template with a biotin moiety attached to triethylene glycol (TEG) spacer at each end and a 28 nt primer whose 5′ end was labeled with Cy5. The sequences of both oligos, respectively, are as follows: /5′BiotinTEG/ATCTTCCTTCAACCAGC T′TACCTTCAACGATTTAGGTTACCTTCAATGTCATGCTCGCGCTGA/3′Bio-TEG/) and /5′Cy5/CAGCGCGAGCATGACATTGAAGGTAACC-3′). The substrate was annealed by mixing both template and primer at a 1:1 molar ratio in TE-100 buffer [50 mM Tris-HCl (pH 8.0), 1 mM EDTA and 100 mM NaCl] and heating at 95 °C for 5 min followed by slow cooling down to room temperature. The annealed product was PAGE purified to >90% purity using 10% non-denaturing poly-acrylamide gel electrophoresis (Invitrogen). The biotin-labeled substrate was incubated with a two-fold molar excess of neutravidin before the primer extension assay. Finally, the substrates were aliquoted and stored at −20 °C.

For the substrates used in cryo-EM, a template strand (5′- CTGCACGAATTA AGCAATTCGTAATCATGGTCATAGCT-3′) was annealed to a primer containing a dideoxycytosine at the 3′ end (5′-AGCTATGACCATGATTACGAATTG[ddC] −3′) to form the P/T substrate. The oligos were mixed in an equimolar ratio in the presence of 20 mM Tris-HCl (pH 7.5) and 25 mM NaCl and annealed by heating at 92 °C for 2 min followed by slow cooling to room temperature. Both oligos were purchased from Sigma Aldrich.

**Primer extension assays.** The primer extension activity assays of Pol κ were performed on either the non-damage substrate in 10 µl total reaction volume.

Briefly, 20 nM non-damage substrate was incubated with the proteins at 30 °C, respectively, in the reaction buffer [40 mM Tris-HCl (pH 7.8), 50 mM NaCl, 0.2 mg/ml BSA, 1 mM DTT, 5 mM MgCl$_2$, 1 mM ATP, 0.1 mM of each deoxynucleotide (dATP, dTTP, dGTP, and dCTP)]. The reactions were terminated with the addition of 10 µl of stop buffer [50 mM EDTA, 95% formamide] followed by heating at 95 °C for 3 min and cooling down on the ice for 2 min. All products were resolved on 15% polyacrylamide gels containing 8 M urea and visualized using Typhoon Trio fluorescence imager (GE Healthcare).

The band intensities in each lane of each of the DNA primer extension assay gels were quantified using the GelAnalyzer software (www.gelanalyzer.com). The amount of DNA synthesis was analyzed using an adaptation of the previously described median analysis method[71] as the intensity of each synthesis band in one lane, that is, any band other than the one corresponding to the unmodified substrate was normalized to the total summed intensity of DNA synthesis in that corresponding lane. The resulting fractional percentages were then summed in a cumulative manner to generate the total percentage of DNA synthesis up to a given number of nucleotides. These cumulative synthesis curves were fitted to smoothing splines and the median number of nucleotides was determined for each lane, that is, the number of synthesized nucleotides up to which 50% of the total DNA synthesis is achieved. Next, this number was divided by the experimental assay time to yield an apparent median DNA synthesis rate which allows to compare the different experimental conditions.

**Cryo-EM grid preparation and data collection.** For the dataset of the Pol κ holoenzyme with wt-PCNA, a 40 µl inject containing 3.75 µM P/T DNA, 3.78 µM Pol κ, 1.5 µM PCNA trimer and 20 µM dTTP was loaded onto a Superdex 200 increase 3.2/300 column (GE Life Sciences) equilibrated with a buffer comprising [25 mM HEPES (pH 7.5), 100 mM potassium acetate, 10 mM calcium chloride, 0.02% NP-40, 0.4 mM biotin and 1 mM DTT]. Three microliters of a fraction corresponding to the first peak (Supplementary Fig. 2, fraction and lane 2) was used. For the dataset of the Pol κ holoenzyme with Ub-PCNA, a 25 µl inject containing 8 µM P/T DNA, 4 µM Pol κ, 4 µM PCNA trimer and 20 µM TTP was loaded onto the same column, equilibrated with a buffer comprising (25 mM HEPES (pH 7.5), 100 mM potassium acetate, 10 mM calcium chloride, 0.02% NP-40, 0.5 mM TCEP). Three microliters of a fraction corresponding to the first peak (Supplementary Fig. 7, fraction and lane 3) was used. For both complexes, UltrAuFoil® R1.2/1.3 Au 300 grids were glow discharged for 5 min at 40 mA on a Quorum Gloqube glow-discharge unit, then covered with a layer of graphene oxide (Sigma) prior to application of the sample. Once the sample was applied to the grid, it was blotted for 3 s and plunge frozen into liquid ethane using a Vitrobot Mark IV (FEI Thermo Fisher), set at 4 °C and 100% humidity. Cryo-EM data for both complexes were collected on a Thermo Fisher Scientific Titan Krios G3 transmission electron microscope at the Midlands Regional Cryo-EM Facility at the Leicester Institute of Structural and Chemical Biology. For both datasets, electron micrographs were recorded using a K3 direct electron detector (Gatan Inc.) at a dose rate of 11 e-/pix/s and a calibrated pixel size of 1.086 Å using EPU 2.3. Data were acquired with a defocus range between −2.0 and −0.8 µm, in 0.3 µm intervals.

**Cryo-EM image processing.** For the dataset of Pol κ with wt-PCNA, data were processed in two parts and combined after polishing. In both parts, pre-processing was performed in Relion-3.1[72] as follows: movie stacks imported in super-resolution mode, then corrected for beam-induced motion and then integrated using Relion's own implementation, using a binning factor of 2. All frames were retained and a patch alignment of 5 × 5 was used. Contrast transfer function (CTF) parameters for each micrograph were estimated by CTFFIND-4.1[73]. Integrated movies were inspected with Relion-3.1 for further image processing (8777 and 2714 movies, with 377 movies common to both – 11,114 distinct movies total). Particle picking was performed in an automated mode using the Laplacian-of-Gaussian (LoG) filter implemented in Relion-3.1. All further image processing was performed in Relion-3.1. Particle extraction was carried out from micrographs using a box size of 300 pixels (pixel size: 1.086 Å/pixel). An initial dataset of $1.1 \times 10^7$ or $2.3 \times 10^6$ particles respectively was cleaned by 2D classification followed by 3D classification with alignment, 3D refinement and several rounds of polishing and per-particle CTF refinement. The data common to both paths were removed from one, before the two partial datasets were joined and further refined. This yielded a 3.4 Å reconstruction of the Pol κ-PCNA-DNA complex comprising 254,040 particles.

Pre-processing of the dataset of Pol κ with Ub-PCNA was performed in much the same way, with $5.3 \times 10^6$ particles initially picked from 3889 movies using 3D reference picking, and extracted using a box size of 300 pixels (pixel size: 1.086 Å/pixel). Two different paths were then taken. To generate the map with visible ubiquitin, a few rounds of 3D classification and refinement and one round of polishing were performed. Then, to remove particles with overrepresented orientations and improve map isotropy, a further round of 2D classification was performed, where each class was limited to 2000 particles. The final map comprised 25,234 particles and had a nominal resolution of 6.4 Å. To generate the map with invisible ubiquitin, a few rounds of 3D classification and refinement and one round of polishing yielded a map comprising 137,301 particles, at a nominal resolution of 3.7 Å. The final half-maps of the reconstruction with wt-PCNA and the reconstruction with Ub-PCNA with visible ubiquitins were used to produce density

modified maps using the Phenix's tool ResolveCryoEM[74]. These maps showed significant improvements in side chain density and overall interpretability.

**Molecular modeling**

*Model building of Pol κ−DNA−wt-PCNA complex.* The X-ray structure of the catalytic domain of human Pol κ bound to P/T DNA and dTTP (PDB ID 2OH2)[30], and the structure of PCNA homotrimer (from PDB ID 6TNZ)[31] were rigid-body fitted into the cryo-EM map. N-clasp residues 21−45 of Pol κ X-ray structure[30] were invisible in the map and were deleted from the model. The upstream 19 base pairs of B-form duplex DNA were built with Chimera[75] and Coot[76] and real-space refined with Coot. The region of Pol κ at the PAD C-terminus (residues 517−534) was built and refined with Coot. The entire model of the Pol κ complex was subjected to real-space refinement in Phenix[44] with the application of secondary structure and stereochemical constraints.

*Model building of the Pol κ−DNA−UbPCNA complex.* The model was built based on the structure of the Pol κ−DNA−wtPCNA complex, which was partitioned into the following sub-structures: Pol κ−DNA complex and PCNA trimer. These structures were individually rigid-body fitted into the cryo-EM map of Pol κ−DNA −UbPCNA with Chimera[75]. Due to the weak density and low map local resolution at the ubiquitin positions, ubiquitin was not included in the final model. The final models were validated using Phenix[44].

**MD simulations**. Simulations were started using two different models of the Pol κ −PCNA complex. The first model (apo1) was generated from the cryo-EM structure of the Pol κ−DNA−wtPCNA complex after removing the DNA. The second model (apo2) was obtained from apo1 with the following steps: the X-ray structure of apo Pol κ (PDB ID 1T94)[31] was aligned to Pol κ PAD domain in apo1 model. Pol κ core domain of apo1 model was then extracted and aligned to the core domain of 1T94, and the loop connecting the core and PAD domains was rebuilt. In both apo1 and apo2 models, some of the disordered residues that could not be resolved in the cryo-EM structure were reconstructed with Modeller[77,78]. The regions of Pol κ spanning residues 1−36 and 225−281, absent in the cryo-EM structure, were not reconstructed because obtaining conformationally converged ensembles for long disordered regions is slow and because there is no evidence that these regions participate in contacts with PCNA. Residues Met225 and Gln281 are in close proximity and were joint by modeling their positions and that of the two bridging residues Gly226 and Leu280. We used the recently developed DES-Amber force field which aims to correctly describe protein−protein interactions[79]. The setup of the simulation closely followed the procedure described in ref. [80]. Briefly, the initial structures were solvated in a dodecahedron box 1 nm away from initial protein atoms. The water model used was TIP4P-D also parametrized in conjunction with the protein force field[79]. $Na^+$ and $Cl^−$ ions were added to simulate a 100 mM NaCl solution. This structure was minimized and run in an NVT ensemble for 2 ns and in an NPT ensemble for additional 2 ns. These simulations had positional restrains of 1000 kJ/mol/nm to all non-hydrogen protein atoms to allow relaxation of the solvent. From these structures, two unrestrained molecular dynamics simulations for each of the two systems were started. Virtual sites in the setup of these simulations were used, which allowed a time step of 4 fs. All simulations were run with Gromacs 2019.4[81–83]. The analysis of the trajectories was carried out with in-house scripts using MDTraj[84]. All four trajectories were concatened into a single trajectory. Then all frames of this trajectory were superimposed onto its first frame and Cartesian coordinates of the backbone atoms were used to calculate the principal components with the implementation in scikit-learn library[85]. Plots were produced with the Matplotlib library[86]. Tools in ENCORE to evaluate the convergence of the simulations[87] were used. For visualization of the trajectories, VMD[88] and Pymol[89] were used.

**NMR spectroscopy**. NMR spectra were acquired from 200 µl samples in standard 3 mm NMR tubes of approximately 158 µM $^{15}$N-labeled ubiquitin (as a control spectrum), and 98 µM $^{15}$N-labeled ubiquitin with either 980 µM of UBZ1 or 980 µM of UBZ2 peptide in a buffer comprising 25 mM HEPES (pH 7.5), 100 mM potassium chloride, 10 mM calcium chloride, 0.5 mM TCEP and 980 µM zinc chloride, containing 10% $D_2O$/90% $H_2O$. Data were acquired at 298 K on a 600 MHz Bruker Avance III spectrometer equipped with a 5 mm, z-gradient TCI cryoprobe. The 2D $^{15}$N-HSQC spectra were recorded using in-house written, standard HSQC experiments with acquisition times of 60 ms for $^{15}$N ($F_1$) and 80 ms $^1$H ($F_2$), using the WATERGATE method to suppress the water signal[90]. All NMR data were processed using Topspin (version 3.5pl7) with linear prediction used to extend the effective acquisition times by two-fold in nitrogen. The HSQC spectra were referenced to water at 298 K, and spectra were analyzed using the NMRFAM Sparky package[91].

The perturbation of the chemical shift of the backbone amide proton and nitrogen signals upon peptide binding can be calculated using the minimum chemical shift procedure[92,93], using the following equation:

$$\min \Delta\delta = \min \left[ \left(^{HN}\Delta_{ppm}\right)^2 + \left(^{N}\Delta_{ppm} * 0.2\right)^2 \right]^{1/2}$$

where $^{HN}\Delta_{ppm}$ and $^{N}\Delta_{ppm}$ are the $^1H_{HN}$ and $^{15}N_{HN}$ respectively. The change in

chemical shift upon peptide binding was mapped onto the NMR structure of ubiquitin (PDB ID 1MRE), so aiding the identification of the binding site.

**Reporting summary**. Further information on research design is available in the Nature Research Reporting Summary linked to this article.

## Data availability

The data that support this study are available from the corresponding authors upon reasonable request. The maps of the Pol κ holoenzymes with wt- and Ub-PCNA have been deposited in the EMBD with accession codes EMD-12601 and EMD-12602, and the atomic models in the Protein Data Bank under accession codes PDB 7NV0 and 7NV1. Additional atomic models used in this study are: Pol κ catalytic domain bound to DNA (PDB 2OH2 []); apo Pol κ (PDB 1T94); PCNA homotrimer (from PDB 6TNZ); Rad18-UBZ/ubiquitin complex (PDB 2MRE); Pol δ holoenzymes (PDB 6TNY and 6S1O); mono-ubiquitylated PCNA (PDB 3L0W and 3TBL).

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

## Acknowledgements

This research was supported by King Abdullah University of Science and Technology through core funding (to S.M.H.) and the Competitive Research Award Grant CRG8 URF/1/4036-01-01 (to S.M.H. and A.D.B.), and by the Wellcome Trust (to A.D.B.). R.C. acknowledges funding from the MINECO (CTQ2016-78636-P) and to AGAUR, (2017 SGR 324). The MD project has been carried out using CSUC resources. We acknowledge The Midlands Regional Cryo-EM Facility at the Leicester Institute of Structural and Chemical Biology (LISCB), major funding from MRC (MC_PC_17136). We thank Christos Savva (LISCB, University of Leicester) for his help in cryo-EM data collection and advice on data processing.

## Author contributions

M.Tehseen purified Pol κ mutants and PCNA, confirmed their activities, and helped in initiating the project; M.Tehseen, M.Takahashi, M.A.S. and V.S.R. optimized and performed the functional assays; S.B. and M.P. prepared Ub-PCNA. C.L. and K.B. prepared the cryo-EM samples; C.L., T.J.R., and A.D.B. analyzed the cryo-EM data; C.L. and A.D.B. built and refined the molecular models. R.C. performed and analyzed the MD simulations. F.W.M. acquired and analyzed the NMR data. S.M.H. and A.D.B. conceived the research and wrote the article. All authors discussed the results and commented on the manuscript.

## Competing interests

The authors declare no competing interests.
