## [Peer Review File · Nature Communications]

Cryo-EM structure of human Pol κ bound to DNA and mono-ubiquitylated PCNAReviewers' Comments:

Reviewer #1:

Remarks to the Author:

In the submitted manuscript, Lancey et al. report two cryo-EM structures of DNA polymerases bound to PCNA and primer-template substrate. The structure of the TLS polymerase Polk-PCNA-DNA complex was resolved to 3.9Å resolution, and that of a stalled Pold-PCNA-DNA complex to 4.7Å resolution. In the Polk-PCNA-DNA structure, the authors observe that Polk docks onto PCNA in a tilted manner, bending the DNA between the polymerase and PCNA. In the stalled Pold structure, the authors argue that the DNA is dislodged from the active site but remains attached to Pold via the thumb domain. The authors also perform MD simulations showing that Polk can sample different orientations in space with respect to bound PCNA. Based on these observations the authors propose a model for polymerase switching during translesion synthesis, either involving a toolbelt mechanism or polymerase dissociation.

The author's work extends numerous previous studies, both from their own lab as well as other labs, that provided structures of PCNA, Polk (with and without DNA), and the Pold holoenzyme with or without PCNA, and of Polk fragments bound to PCNA. The novelty of the authors' work lies in obtaining a structure of Polk in complex with PCNA, which was previously only obtained for Pold. However, one major concern is that the resolution of the author's structures is with 3.9Å and 4.7Å relatively low and does not provide sufficient detail to unambiguously place side chains into the density, yet the authors provide no biochemical validation of their interpretation of the structural model. Moreover, the authors' findings do not resolve whether polymerase switching occurs via the toolbelt model or by polymerase dissociation, although the authors would be in a perfect position to test these possibilities directly. Overall, the novel functional and mechanistic insights gained by the current structures are limited and no additional data are provided to validate and test the proposed models. To publish this manuscript, such data would be essential.

Major concerns:

- 1) The authors suggest that Pold and Polk can be simultaneously bound to the same PCNA ring without steric clashing, which predicts that a complex of Polk-Pold-PCNA-DNA could be reconstituted. This should be explicitly tested biochemically by the authors to support their model.
- 2) It remains unclear how exactly DNA is transferred from Pold to Polk. The stalled Pold-PCNA-DNA structure shows that, although DNA may be dislodged from the active site, it remains stably bound to the complex. Is there an affinity difference for Pold binding to DNA in the two different complexes to support the authors' model?
- 3) Suppl. Fig. 1: The authors use P/T templates that significantly differ in length and base composition for the biochemical experiments, preventing direct comparison of the experimental results. It would have made more sense to use the same P/T and only modify the site that is abasic. For 1b and 1c, it is unclear why RFC was added to experiments in 1b but not in 1c, and why protein concentrations differ between the experiments.
- 4) Suppl. Fig. 3: The Polk-PCNA-DNA particles appear to adopt a preferred orientation, yet the anisotropic resolution of the structure is not analysed. The author should include such an analysis.
- 5) Suppl. Fig. 4: The scheme indicates that the authors did not refine the second 3D class (113,560 particles). Why? This class looks similar to class 3 (which was refined) and also appears to correspond to Polk-PCNA-DNA.
- 6) The authors show little density for protein side chains (with the exception of Figs. 2c 3e, which are

both hard to interpret). Images of density overlaid onto the atomic model for key regions should be included in the supplemental material to allow the reader to judge the local map quality and resolution. For example: PCNA-Polk interface, the Polk active site, the DNA-contacting region of PCNA.

7) The authors mention several residues involved in the interaction between PCNA, PAD, and the PIP-box, yet none of these interactions is validated biochemically. Given the limited resolution of the structure, it would be important to mutate residues directly involved in the interaction (Q526, R527, S528 and I529 in Polk) and test their effects on holocomplex assembly.

8) Fig. 3e: The density around the individual side chains is hard to interpret as it is shown without context of the surrounding density. The authors should show a better image including extended density for PCNA, as well as the DNA.

9) Fig. 4b: In the simulated structure shown, the interaction between Polk and PCNA appears limited to the PIP-box. What prevents Polk from dissociating from PCNA? What's the affinity for the PIP-box/PCNA interaction?

10) Fig. 6: The authors describe conformational changes in the stalled Pold-PCNA-DNA complex compared to their previous replicating state. However, the changes, with exception of the finger domain movement, are relatively minor. Some of the proposed changes appear to involve movements involving shorter distances than the resolution of 4.7Å. It remains thus unclear whether some of these minor changes are physiologically relevant or correspond to breathing of the complex. These slight movements also indicate that the structures could benefit from multi-body refinement. Does this approach improve the resolution? What are the principal components of movement in the stalled complex and how do they compare to the replicating one?

11) Suppl. Table 1: The model resolution was determined at 0.143 FSC. However, it is common practice to evaluate model resolution by model to map FSC at a cutoff of 0.5 as using the 0.143 cutoff results in overestimation of model resolution. These map/model FSC curves should also be included in a supplemental figure. Moreover, what are the ligands modelled into the structures? The clashscore of the stalled Pold-PCNA-DNA complex is with 12.36 also quite high, as are the Ramachandran outliers of >1%, suggesting the model was overrefined. The authors should improve the model.

Minor concerns:

1) Page 3: The sentence "Recent work showed that, in yeast, the lagging strand Pold is responsible for lesion bypass on both leading- and lagging strands" requires a reference.

2) Page 4: PAD is not defined.

3) Fig. 1b: It would be helpful if the authors would label the thumb and palm domains.

4) Page 4: The sentence "In the apo enzyme, the PAD is positioned under the palm domain in two alternate positions..." is confusing as the PAD is also positioned under the palm domain in the DNA bound complex, and it is unclear which state the alternate positions refer to.

5) Suppl. Fig. 2: It would be helpful to show which lanes from the gel filtration were pooled and used for cryoEM.

6) Page 13 and Fig. 5i: There are discrepancies between the text and figure with respect to the number of bp that the DNA extends upon exiting the PCNA hole.

Reviewer #2:

Remarks to the Author:

In this manuscript, Lancey et al. report cryo-EM structures of human Pol kappa and Pol delta, both bound to a template and primer DNA and the PCNA clamp. The group has recently reported the human Pol delta structure bound to T/P and PCNA, but the T/P is partially disengaged from the polymerase site in the current structure at 4.7 Å resolution. The Pol kappa structure bound to T/P was also previously reported, but not with the PCNA clamp, as seen in the current structure at a nominal resolution of 3.9 Å. The quality of the Pol kappa complex is low and anisotropic and needs to be experimentally improved. Based on the structures and an MD simulation, the authors propose a polymerase switch mode in which the Pol delta may remain attached when TLS Pol is transiently recruited to bypass a lesion. The work contains some new insights, such as the Polk PCNA interaction and the backtracking of the DNA substrate in PolD. However, the novelty is limited and this, combined with several major concerns on the quality of the data stated below, seems to suggest that the manuscript in its current form may be unsuitable for publication in this journal.

Major concerns

1) Although the average resolution is at 3.9 Å, the quality of the cryo-EM map of the Pol kappa is low and anisotropic. There appears to be severe preferred orientation problem (based on presented 2D averages), consistent with very early drop in the FSC curve, and distorted density feature, and the lack of DNA major and minor groove definition. I am not sure if the DNA bending is real (maybe) or a result of density distortion due to the preferred particle orientation. I don't feel the current map is good enough for publication, and the authors should strive to overcome the problem to improve the map quality.

2) It is an overstatement in the MS title that the authors have determined the structure of the Pol kappa holoenzyme, because the C-terminal third of the Polk is not observed. But more seriously, the SDS-PAGE shows Polk may be partially degraded, which may explain the lack of C-terminal 1/3 of the structure (Fig. S2b). It is suggested that the CT 1/3 may be flexible; but no experimental evidence is presented. SAXS analysis of the full length versus the CT 1/3 truncated enzyme would provide some evidence.

3) The authors attempt to link up the two separate structures of PolD and PolK into a Pol swap mechanism by the MD simulation. But this is speculative. A relatively straightforward experiment such as negative stain EM or 2D class averages of the PolD/PolK mixture should be carried out to show if the two enzymes can simultaneously associate (one of them flexibly) with the same PCNA clamp.

The MS has numerous typos; a few are listed below.

1) page 9 line 2, "primer" should be "template".

2) page 10 2nd paragraph (MD simulations). the authors should collect a small dataset of a mixture of Pol k and PCNA (without DNA) to show in 2D averages this possibility.

3) page 13 2nd paragraph line 6, the length in the Fig. 5a solid box is 26 bp, the label in the Fig. 5i is 25 bp, also line 8 states "3bp".

4) page 21 2nd paragraph line 9 is not consistent with the label in SI Fig. 1a.

5) page 23 2nd paragraph line 5, "xx" should be replaced by "2"?

6) page 25 1st paragraph last sentence, "several" – actual number. too many rounds of polishing may introduce the artifacts. State resolution before and after polishing.

- 7) page 26 line 1 and 2, "κ" should be "δ".
- 8) Page 38 figure legend 1, line 1, "(a)" need to be deleted.
- 9) Page 39 figure legend 2, line 5, missing a full stop.
- 10) Page 41, line 8, figure legend 4, panel b, the label "CORE" should be consistent throughout the text.
- 11) Page 42 figure legend 5, The labels in panels a-i should be reordered, the label "THUMB" in panel g should be consistent through the text, also the labeled DNA length in panel i is not consistent with panel a.
- 12) Page 44 figure legend 6, line 1, "(a)" needs to be deleted; panel 6a left, the two Pol D structures should be shown in different colors.
- 13) Page 46 figure legend 7, last line, missing a full stop.
- 14) Fig S1 panel a, the middle line in the diagram of abasic substrate should be deleted.
- 15) Fig S6 legend, no title.
- 16) Table S1, the poor rotamers and disallowed seems mixed, not consistent with the PDB report summary (page 2).

REVIEWER COMMENTS

Reviewer #1 (Remarks to the Author):

In the submitted manuscript, Lancey et al. report two cryo-EM structures of DNA polymerases bound to PCNA and primer-template substrate. The structure of the TLS polymerase Polk-PCNA-DNA complex was resolved to 3.9Å resolution, and that of a stalled Pold-PCNA-DNA complex to 4.7Å resolution. In the Polk-PCNA-DNA structure, the authors observe that Polk docks onto PCNA in a tilted manner, bending the DNA between the polymerase and PCNA. In the stalled Pold structure, the authors argue that the DNA is dislodged from the active but remains attached to Pold via the thumb domain. The authors also perform MD simulations showing that Polk can sample different orientations in space with respect to bound PCNA. Based on these observations the authors propose a model for polymerase switching during translesion synthesis, either involving a toolbelt mechanism or polymerase dissociation.

The author's work extends numerous previous studies, both from their own lab as well as other labs, that provided structures of PCNA, Polk (with and without DNA), and the Pold holoenzyme with or without PCNA, and of Polk fragments bound to PCNA. The novelty of the authors' work lies in obtaining a structure of Polk in complex with PCNA, which was previously only obtained for Pold. However, one major concern is that the resolution of the author's structures is with 3.9Å and 4.7Å relatively low and does not provide sufficient detail to unambiguously place side chains into the density, yet the authors provide no biochemical validation of their interpretation of the structural model. Moreover, the authors' findings do not resolve whether polymerase switching occurs via the toolbelt model or by polymerase dissociation, although the authors would be in a perfect position to test these possibilities directly. Overall, the novel functional and mechanistic insights gained by the current structures are limited and no additional data are provided to validate and test the proposed models. To publish this manuscript, such data would be essential.

We thank the referee for their suggestions.

We agree with the referee that the suggested experiments are required to validate our hypotheses. We therefore carried out further biochemical and structural work to test the mechanism of polymerase switching (please see replies to Major Concerns). Because our findings could not unambiguously resolve whether Polk and Pold exchange via the toolbelt or sequential model, and because in the meantime we generated new cryo-EM structures, the focus of the manuscript has slightly changed. We now present, together with an improved cryo-EM reconstruction of Polk bound to DNA and wild type PCNA at 3.4 Å resolution, a structure of Polk bound to DNA and mono-ubiquitylated PCNA (UbPCNA) between 3.7 Å and 6.4 Å resolution. PCNA mono-ubiquitylation at K164 is a key modification that regulates the access of Polk, and TLS polymerases in general, to sites of DNA damage. The structural basis of the interaction of TLS polymerases with UbPCNA is poorly understood, as structural information is limited to UbPCNA in isolation (Freudenthal et al, NSMB, 2010; Tsutakawa et al, PNAS, 2011; Zhang et al, Cell Cycle, 2012.; Hibbert et al, JBC, 2012). In summary, our new structures show that: the position of Polk relative to UbPCNA in the complex is entirely dictated by the internal PIP-box interaction and by the interaction of Polk with DNA, (ii) the ubiquitins extend radially away from PCNA and are partly flexible, with their hydrophobic patch mostly accessible to interact with the two Polk Ubiquitin Binding Zing Fingers (UBZs), and (iii) the C-terminal region of Polk encompassing the two UBZs is flexible and is not rigidified upon binding to UbPCNA, implying that the interaction between the UBZs and ubiquitin is transient and/or comprises various orientations of the ubiquitin moieties.

To validate and expand our structural findings, we performed additional functional studies. We expressed and purified six different Polk variants containing mutations at the PCNA interface and in the UBZs, and tested their activities in primer extension assays with wt- or

UbPCNA. Our data, shown in Figure 5 of the current manuscript, suggest that the PCNA-enhanced activity of Polk is controlled primarily by the internal PIP-box, which overrides the secondary UBZ/ubiquitin interaction. When the PIP-box interaction is lost, the UBZ/ubiquitin interaction becomes significant in retaining Polk on PCNA and preventing Polk detachment from the DNA template.

We believe that these findings significantly expand our understanding of the molecular basis for the PCNA-directed recruitment of Polk to DNA. Because of the domain conservation of Y-family polymerases Polk, eta, and iota, the structural features observed in Polk complexes are likely general and may apply to the corresponding complexes of Pol eta and iota.

Given the new focus of the manuscript, and because of the limited novel information contained in the 4.7 Å resolution structure of stalled Pold, as well as the uncertainties regarding Pold and Polk exchange, we removed the stalled Pold structure from the manuscript, and we adjusted the model for polymerase exchange assigning equal probabilities to the toolbelt or sequential model (Figure 6).

Major concerns:

1) The authors suggest that Pold and Polk can be simultaneously bound to the same PCNA ring without steric clashing, which predicts that a complex of Polk-Pold-PCNA-DNA could be reconstituted. This should be explicitly tested biochemically by the authors to support their model.

We attempted to reconstitute a Polk-Pold-PCNA-DNA complex biochemically and visualize it by cryo-EM. Polk, Pold, PCNA, P/T DNA and dTTP were mixed and run in a micro-SEC column (Figure 1a below). The eluted peak was broad but early peak fractions showed the presence of all proteins (Figure 1b). One of these fractions was vitrified and imaged by cryo-EM (Figure 1c). Most of the 2D class averages clearly show the presence of either Pold-DNA-PCNA complex or Polk-DNA-PCNA complex, while in some 2D classes PCNA is surrounded by a fuzzy density which could not be clearly assigned. While these observations may support the sequential model for polymerase exchange, we feel that they are not conclusive, as a short-lived toolbelt with both polymerases bound to PCNA may exist but may not be resolved in the experiment.

1) Cryo-EM of Pol κ /Pol δ /DNA/PCNA mixture separated by micro-SEC

2) Cryo-EM of Pol κ_{47-870} /Pol δ /DNA/PCNA mixture prepared by direct mixing

In order to capture a transient state of the complex with both Pold and Polk bound to PCNA but with Polk disengaged from P/T DNA, we expressed and purified a Polk variant lacking 46 residues at the N-terminus. Such N-terminal deletion has been previously shown to decrease the DNA binding affinity of Polk by 4-fold (Lone, Mol Cell, 2007). This Polk mutant was mixed with Pold, PCNA, P/T DNA and dTTP, and the mixture vitrified and imaged by cryo-EM (Figure 2). 2D class averages only showed the presence of Pold complex, with no indication of Polk bound. However, the high conformational flexibility of the apo form of Polk, predicted by our MD simulations and suggested by previous structural studies (Uljon, Structure, 2004), may have resulted in density averaging, making Polk invisible in the 2D classes.

Overall, due to the uncertainties inherent in these experiments, we believe that the mechanism of polymerase exchange is still not resolved. We therefore assigned equal probabilities to the sequential and toolbelt models in the current manuscript model figure (Figure 6).

2) It remains unclear how exactly DNA is transferred from Pold to Polk. The stalled Pold-PCNA-DNA structure shows that, although DNA may be dislodged from the active site, it remains stably bound to the complex. Is there an affinity difference for Pold binding to DNA in the two different complexes to support the authors' model?

A decrease in the DNA binding affinity for stalled Pold is expected, because DNA is released from the active site and is held bound to the complex only through the thumb domain. However, we agree with the referee that the mechanism of DNA handoff is not resolved (please see comments above).

3) Suppl. Fig. 1: The authors use P/T templates that significantly differ in length and base composition for the biochemical experiments, preventing direct comparison of the experimental results. It would have made more sense to use the same P/T and only modify the site that is abasic. For 1b and 1c, it is unclear why RFC was added to experiments in 1b but not in 1c, and why protein concentrations differ between the experiments.

Because of the new focus of the manuscript, the experiment shown in Supp. Fig.1 has been removed. A new set of biochemical experiments on Polk activity is now shown in Figure 5.

4) Suppl. Fig. 3: The Polk-PCNA-DNA particles appear to adopt a preferred orientation, yet the anisotropic resolution of the structure is not analysed. The author should include such an analysis.

A new, larger dataset of Polk-PCNA-DNA complex was acquired and merged with the original dataset (Supplementary Figures 2 and 3). This improved the map resolution from 3.9 to 3.4 Å, and reduced the moderate anisotropy of the original map. The final map was subjected to Phenix Density Modification, which further improved side chain density and overall map interpretability. An anisotropy analysis is now shown in Supplementary Figure 2. We are convinced that the residual map anisotropy does not affect any of the conclusions drawn in the manuscript. The map and model are now made available to the referee for a visual inspection.

5) Suppl. Fig. 4: The scheme indicates that the authors did not refine the second 3D class (113,560 particles). Why? This class looks similar to class 3 (which was refined) and also appears to correspond to Polk-PCNA-DNA.

The second 3D class of the original dataset (now shown in the second path of data processing in Supplementary Figure 3) presents distortions in the Polk region, and refinement did not lead to a meaningful reconstruction.

6) The authors show little density for protein side chains (with the exception of Figs. 2c 3e, which are both hard to interpret). Images of density overlaid onto the atomic model for key regions should be included in the supplemental material to allow the reader to judge the local map quality and resolution. For example: PCNA-Polk interface, the Polk active site, the DNA-contacting region of PCNA.

Several images of density overlaid to the atomic model (including Polk PIP-box, Polk active site, P/T DNA and PAD aQ helix), documenting the quality of the map and model, are now included in Figure 2a of the manuscript. Density at the DNA-contacting region of PCNA is weak (as is DNA density for the bases threading PCNA) and is not shown. This feature is now discussed clearly in the Results section (Page 9).

7) The authors mention several residues involved in the interaction between PCNA, PAD, and the PIP-box, yet none of these interactions is validated biochemically. Given the limited resolution of the structure, it would be important to mutate residues directly involved in the interaction (Q526, R527, S528 and I529 in Polk) and test their effects on holocomplex assembly.

The Polk/PCNA interface has now been validated biochemically using Polk mutants in primer/extension assays. Three Polk variants containing mutations in the inverting helix or the PIP-box (including the mutant suggested by the referee) were expressed and purified and used in these assays (Figure 5).

8) Fig. 3e: The density around the individual side chains is hard to interpret as it is shown without context of the surrounding density. The authors should show a better image including extended density for PCNA, as well as the DNA.

Please see the comments above.

9) Fig. 4b: In the simulated structure shown, the interaction between Polk and PCNA appears limited to the PIP-box. What prevents Polk from dissociating from PCNA? What's the affinity for the PIP-box/PCNA interaction?

This has been now clarified in the MD result section with the following sentence (Page 10): "Across all simulations, Polk maintains the interaction with PCNA via the three PIP-box residues Ile529, Phe532 and Leu533 inserted into the canonical hydrophobic cleft, and Gln526 bound to the Q-pocket".

The affinity of the Polk internal PIP-box for PCNA has not been measured, but based on the PIP-box sequence conservation and previous affinity measurements of PIP-box/PCNA interactions for other Y-family TLS polymerases (Hishiki, JBC, 2009), it is expected that the interaction is in the low micromolar range.

10) Fig. 6: The authors describe conformational changes in the stalled Pold-PCNA-DNA complex compared to their previous replicating state. However, the changes, with exception of the finger domain movement, are relatively minor. Some of the proposed changes appear to involve movements involving shorter distances than the resolution of 4.7Å. It remains thus unclear whether some of these minor changes are physiologically relevant or correspond to breathing of the complex. These slight movements also indicate that the structures could benefit from multi-body refinement. Does this approach improve the resolution? What are the principal components of movement in the stalled complex and how do they compare to the replicating one?

Because of the reasons explained above, the stalled Pold structure has been removed from the current manuscript.

11) Suppl. Table 1: The model resolution was determined at 0.143 FSC. However, it is common practice to evaluate model resolution by model to map FSC at a cutoff of 0.5 as using the 0.143 cutoff results in overestimation of model resolution. These map/model FSC curves should also be included in a supplemental figure. Moreover, what are the ligands modelled into the structures? The clashscore of the stalled Polδ-PCNA-DNA complex is with 12.36 also quite high, as are the Ramachandran outliers of >1%, suggesting the model was overrefined. The authors should improve the model.

Map to model plots are now included in Supplementary Figures 2 and 7. The ligand modelled in the Polκ complexes is dTTP in the active site.

Minor concerns:

1) Page 3: The sentence “Recent work showed that, in yeast, the lagging strand Polδ is responsible for lesion bypass on both leading- and lagging strands” requires a reference. This sentence is no longer present in the manuscript.

2) Page 4: PAD is not defined. This has been addressed.

3) Fig. 1b: It would be helpful if the authors would label the thumb and palm domains. Polκ subdomains are now coloured differently in the figure and labelled.

4) Page 4: The sentence “In the apo enzyme, the PAD is positioned under the palm domain in two alternate positions...” is confusing as the PAD is also positioned under the palm domain in the DNA bound complex, and it is unclear which state the alternate positions refer to.

The sentence has now been reworded: “The apo-enzyme was crystallized with the PAD positioned under the palm domain in two alternate positions, while in the DNA-bound structure the PAD is docked in the major groove”

5) Suppl. Fig. 2: It would be helpful to show which lanes from the gel filtration were pooled and used for cryoEM.

The gel lanes used for cryo-EM are now indicated in Supplementary Figure 1 and 6.

6) Page 13 and Fig. 5i: There are discrepancies between the text and figure with respect to the number of bp that the DNA extends upon exiting the PCNA hole.

This figure has been removed from the manuscript (please see comments above).

Reviewer #2 (Remarks to the Author):

In this manuscript, Lancey et al. report cryo-EM structures of human Polκ and Polδ, both bound to a template and primer DNA and the PCNA clamp. The group has recently reported the human Polδ structure bound to T/P and PCNA, but the T/P is partially disengaged from the polymerase site in the current structure at 4.7 Å resolution. The Polκ structure bound to T/P was also previously reported, but not with the PCNA clamp, as seen in the current structure at a nominal resolution of 3.9 Å. The quality of the Polκ complex is low and anisotropic and needs to be experimentally improved. Based on the structures and an MD simulation, the authors propose a polymerase switch mode in which the Polδ may remain attached when TLS Pol is transiently recruited to bypass a lesion. The work contains some new insights, such as the Polκ PCNA interaction and the backtracking of the DNA substrate in Polδ. However, the novelty is limited and this,

combined with several major concerns on the quality of the data stated below, seems to suggest that the manuscript in its current form may be unsuitable for publication in this journal.

We thank the referee for their useful comments.

This referee's concerns about the mechanism of polymerase exchange are very similar to those from the first referee, and these have been addressed above. We invite this referee to consider our replies to the first referee, and in particular the additional structural and biochemical work we carried out to test the toolbelt model for polymerase exchange. Due to ambiguity of our results concerning this mechanism, and because in the meantime we generated new cryo-EM structures, we slightly changed the focus of the manuscript. In the current manuscript we present, along with an improved cryo-EM reconstruction of Polk bound to DNA and wild type PCNA (determined at 3.4 Å resolution), a structure of Polk bound to DNA and mono-ubiquitylated PCNA (UbPCNA) determined between 3.7 and 6.4 Å resolution. In addition, we present biochemical assays to validate and expand our structural findings. In our reply to the first referee, we explained the content and the functional significance of these new findings.

Major concerns

1) Although the average resolution is at 3.9 Å, the quality of the cryo-EM map of the Pol kappa is low and anisotropic. There appears to be severe preferred orientation problem (based on presented 2D averages), consistent with very early drop in the FSC curve, and distorted density feature, and the lack of DNA major and minor groove definition. I am not sure if the DNA bending is real (maybe) or a result of density distortion due to the preferred particle orientation. I don't feel the current map is good enough for publication, and the authors should strive to overcome the problem to improve the map quality.

A new, larger dataset of Polk-PCNA-DNA complex was acquired and merged with the original dataset (Supplementary Figure 2 and 3 in the current manuscript). This improved the map resolution from 3.9 to 3.4 Å, and reduced the moderate anisotropy of the original map. The final half-maps of the reconstruction were used to produce a density modified map using the Phenix's tool ResolveCryoEM (Terwilliger et al., 2020). The modified map showed significant improvements in side chain density and overall interpretability (and an improvement in resolution to 3.26 Å according to the $FSC_{ref}=0.5$ criterion). The new FSC curve (calculated prior to density modification, Supplementary Figure 2d) no longer shows an early drop. The new map does not show any significant density distortion (Figure 1c-d, Figure 2a, Supplementary Figure 2f). The map region around the DNA (Figure 2a) shows an excellent definition of the major and minor grooves (apart for the bases threading PCNA, a feature that is now discussed clearly on Page 9 of the Results section). We carried out an anisotropy analysis with 3DFSC (Tan et al, 2017) on the map prior to density modification (Supplementary Figure 2c). We are convinced that the residual anisotropy in the map does not affect any of the conclusions drawn in the manuscript. The map and model are now made available to the referee for a visual inspection.

2) It is an overstatement in the MS title that the authors have determined the structure of the Pol kappa holoenzyme, because the C-terminal third of the Polk is not observed. But more seriously, the SDS-PAGE shows Polk may be partially degraded, which may explain the lack of C-terminal 1/3 of the structure (Fig. S2b). It is suggested that the CT 1/3 may be flexible; but no experimental evidence is presented. SAXS analysis of the full length versus the CT 1/3 truncated enzyme would provide some evidence.

The word holoenzyme is no longer present in the current manuscript's title. A disorder prediction of Polk full sequence is now included in Figure 1a, clearly showing that, while Polk core is predicted to be well ordered, the C-terminal domain is predicted to be highly flexible

(with the exception of the two UBZs), and this agrees with the fact that the C-terminal domain is not observed in the cryo-EM map. The precise definition of the short regions in Polk core predicted to be disordered and not observed in the cryo-EM map or the X-ray structure of Polk core (Lone, Mol Cell, 2007) (e.g., loop in the palm domain spanning residues 225-281) suggests that the disorder prediction is reliable. The micro-SEC fraction used for cryo-grid preparation is now labelled on the SDS-PAGE gel (Supplementary Figure 1b), confirming that Polk is intact and not degraded.

Flexibility of the C-terminal region has been previously observed experimentally for Pol eta (Powers et al, NAR, 2018), and appears as a common characteristic of eukaryotic Y-family polymerases (Ohmori et al, Advances in protein chemistry and structural biology, 2008). Given these premises, we do not feel that a SAXS analysis on Polk is essential for the scope of the manuscript.

3) The authors attempt to link up the two separate structures of PolD and PolK into a Pol swap mechanism by the MD simulation. But this is speculative. A relatively straightforward experiment such as negative stain EM or 2D class averages of the PolD/PolK mixture should be carried out to show if the two enzymes can simultaneously associate (one of them flexibly) with the same PCNA clamp.

Following this referee's suggestion, cryo-EM experiments of PolD/PolK/PCNA/DNA mixtures have been carried out. Please see the reply to the first referee.

The MS has numerous typos; a few are listed below.

1) page 9 line 2, "primer" should be "template".

The terminal A is in the primer strand, not the template strand.

2) page 10 2nd paragraph (MD simulations). the authors should collect a small dataset of a mixture of Pol k and PCNA (without DNA) to show in 2D averages this possibility.

We collected a small cryo-EM dataset of a mixture Pol k and PCNA, and processed the data. 2D classes showed well defined PCNA rings with fuzzy density at one ring side. In the figure below, two representative 3D classes of the complex are depicted, showing the PCNA ring with residual density adjacent to one protomer, which presumably arises from flexibly tethered Pol k. This is in agreement with our MD simulations predicting high flexibility of Pol k bound to PCNA in the absence of DNA.

3) page 13 2nd paragraph line 6, the length in the Fig. 5a solid box is 26 bp, the label in the Fig. 5i is 25 bp, also line 8 states "3bp".

This figure is no longer present in the manuscript.

4) page 21 2nd paragraph line 9 is not consistent with the label in SI Fig. 1a.

This Supplementary Figure is no longer present in the manuscript (please see comments to Referee 1)

5) page 23 2nd paragraph line 5, “xx” should be replaced by “2”?

This has been corrected.

6) page 25 1st paragraph last sentence, “several” – actual number. too many rounds of polishing may introduce the artifacts. State resolution before and after polishing.

The number of polishing cycles (3) is now included in the methods section.

7) page 26 line 1 and 2, “κ” should be “δ”.

Pol d structure is no longer included in the manuscript.

8) Page 38 figure legend 1, line 1, “(a)” need to be deleted.

This has been addressed.

9) Page 39 figure legend 2, line 5, missing a full stop.

Figure legends in the current manuscript have been re-written.

10) Page 41, line 8, figure legend 4, panel b, the label “CORE” should be consistent throughout the text.

This has been addressed.

11) Page 42 figure legend 5, The labels in panels a-i should be reordered, the label “THUMB” in panel g should be consistent through the text, also the labeled DNA length in panel i is not consistent with panel a.

This figure has been removed from the manuscript.

12) Page 44 figure legend 6, line 1, “(a)” needs to be deleted; panel 6a left, the two Pol D structures should be shown in different colors.

This figure has been removed from the manuscript.

13) Page 46 figure legend 7, last line, missing a full stop.

This is no longer present in the manuscript.

14) Fig S1 panel a, the middle line in the diagram of abasic substrate should be deleted.

Because of the new focus of the manuscript, the experiment shown in Supp. Fig.1 has been removed. A new set of biochemical experiments on Polk activity is now shown in Figure 5.

15) Fig S6 legend, no title.

This has been addressed.

16) Table S1, the poor rotamers and disallowed seems mixed, not consistent with the PDB report summary (page 2).

A new Table S1 is now presented.

Reviewers' Comments:

Reviewer #1:

Remarks to the Author:

In this revision, Lancey et al. describe two cryo-EM structures of human Pol k in complex with DNA and non-ubiquitinated or mono-ubiquitinated PCNA. The structure of the non-ubiquitinated PCNA complex is similar to that described in the original submission, although the resolution has been slightly improved. The structure of the complex containing ubiquitinated PCNA is new and replaces that of a stalled Pol d-DNA-PCNA complex. Comparison of both Pol k structures shows that Pol k, PCNA, and DNA adopt similar conformations, with the ubiquitins being partially flexible. Although the ubiquitin-binding domains of Pol k are not resolved in either structure, biochemical analysis of DNA synthesis by wild type and various Pol k mutants shows that Pol k ubiquitin binding becomes important when the Pol k PIP-box interaction with PCNA is impaired. While the novelty of the biological insights is still somewhat limited, the modifications and the new data added to the revision substantially improve the manuscript. However, the following issues should be addressed before the work could be accepted for publication.

- 1) The authors state in their response that the map files have been provided to the reviewers, yet these files are not available. The authors should submit these files as UCSF chimera session files or as ccp4 map files.
- 2) Fig. 5: The differences in primer extension for some of the Pol k mutants are relatively subtle and the figure could benefit from quantitation of the synthetic Pol k activity in the different conditions.
- 3) Fig. S8: The figure legend refers to the Pol k-DNA-wt PCNA complex although the Pol k-DNA-Ub-PCNA complex is shown.
- 4) Fig. S10: The lanes show unequal loading of the various Pol k mutants. The authors should state in the figure legend whether this was intentional or not. If the amount was supposed to be similar, how do the inaccuracies in protein concentration affect the interpretation of the biochemical assays in Fig. 5?
- 5) The manuscript submission contains unk files, but there is no description of what these files are and what software can open these files.
- 6) The authors submitted two pdb files as related manuscript file. It is not clear which structure each of the file corresponds to since no explanation is provided.

Reviewer #2:

Remarks to the Author:

In the revised manuscript, Lancey et al. have improved their cryo-EM map of the human Polk-DNA-PCNA complex to 3.4 Å resolution and added a low-resolution map (6.4 Å) with Polk with monoubiquitylated PCNA. They have also removed the poorly defined stalled PolD-DNA-PCNA map. The primer extension assays now suggests that the internal PIP-box plays a more important role than the two UBZs motifs in the PCNA-enhanced activity of Polk. These are good changes that have significantly improved the manuscript. However, we still have concerns that need to be addressed.

Major concerns:

- 1) Fig. 6b. The discussion about the "sequential model" is inadequate. The authors have tried and failed to obtain the Polk-PolD-PCNA-DNA super complex either with direct mixing or by SEC column, and 2D EM averages also failed to capture any complex formation. Those results should be presented

as supplemental figure and discussed, as they are in support of or at least consistent with the sequential model.

2) Fig. 6a. A novelty of the work could be Ub interaction with UBZ1. But the experimental support is inadequate. Similarity to RAD18 UBZ interaction with Ub is OK but a bit handwaving (Fig. 5e). As Polk UBZ1 and UBZ2 are similar in sequence and both could in principle bind to Ub R42 (Fig. 5e). The primer extension assay used combinatorial mutations in both PIP and UBZ1 or UBZ2 and the effects were not clearcut. We will need a clear effect in mutation in UBZ1 but not in UBZ2 to make Fig. 6a convincing. Alternatively, an in vitro binding between UBZ1 domain and Ub but not between UBZ2 and Ub would also work.

A few typos are listed below:

- 1) page 22 line 2, "human" is repetitive.
- 2) page 24 2nd paragraph line 4, the "broth" can be replaced by "media"; the "DNase" throughout the text should be replaced with a formal one "DNase".
- 3) page 25 1st paragraph line 13, missed a preposition after the "eluted".
- 4) page 28 2nd paragraph, "TTP" should be "dTTP", and the formal abbreviation of "K-Ac" should be "KOAc or Kac".
- 5) page 33 4th paragraph line 2, "EMBD" should be "EMDB".
- 6) Page 47 figure legend 5 line 4, there is an extra vertical line following Pol k.
- 7) Fig S6 panel b, missing marker unit "kDa"
- 8) Table S1 1st line 2nd column, "PDD" should be "PDB".

REVIEWER COMMENTS

Reviewer #1 (Remarks to the Author):

In this revision, Lancey et al. describe two cryo-EM structures of human Pol k in complex with DNA and non-ubiquitinated or mono-ubiquitinated PCNA. The structure of the non-ubiquitinated PCNA complex is similar to that described in the original submission, although the resolution has been slightly improved. The structure of the complex containing ubiquitinated PCNA is new and replaces that of a stalled Pol d-DNA-PCNA complex. Comparison of both Pol k structures shows that Pol k, PCNA, and DNA adopt similar conformations, with the ubiquitins being partially flexible. Although the ubiquitin-binding domains of Pol k are not resolved in either structure, biochemical analysis of DNA synthesis by wild type and various Pol k mutants shows that Pol k ubiquitin binding becomes important when the Pol k PIP-box interaction with PCNA is impaired. While the novelty of the biological insights is still somewhat limited, the modifications and the new data added to the revision substantially improve the manuscript. However, the following issues should be addressed before the work could be accepted for publication.

We thank the reviewer for their positive comments. The remaining issues have been addressed as detailed below.

1) The authors state in their response that the map files have been provided to the reviewers, yet these files are not available. The authors should submit these files as UCSF chimera session files or as ccp4 map files.

The map files have now been submitted in ccp4 format.

2) Fig. 5: The differences in primer extension for some of the Pol k mutants are relatively subtle and the figure could benefit from quantitation of the synthetic Pol k activity in the different conditions.

Quantitation of the synthetic activity of Pol k mutants has been carried out and presented in Figure 5e and Supplementary Table 2 of the new manuscript. A detailed explanation of such quantitation is provided in the Materials and Methods section.

3) Fig. S8: The figure legend refers to the Pol k-DNA-wt PCNA complex although the Pol k-DNA-Ub-PCNA complex is shown.

This has been corrected.

4) Fig. S10: The lanes show unequal loading of the various Pol k mutants. The authors should state in the figure legend whether this was intentional or not. If the amount was supposed to be similar, how do the inaccuracies in protein concentration affect the interpretation of the biochemical assays in Fig. 5?

We thank the reviewer for this comment. Pol k and different mutants were loaded by volume, which made them look different. However, amounts were measured correctly and calculated in the assays to ensure that we use similar concentrations. We re-ran the gel and used an equal amount for all proteins (Supplementary Figure 10 of the new manuscript).

5) The manuscript submission contains unk files, but there is no description of what these files are and what software can open these files.

This has now been addressed.

6) The authors submitted two pdb files as related manuscript file. It is not clear which structure each of the file corresponds to since no explanation is provided.

Descriptions of the PDB files are now provided. PDB and map codes are provided in the Author Information section and Supplementary Table 1.

Reviewer #2 (Remarks to the Author):

In the revised manuscript, Lancey et al. have improved their cryo-EM map of the human Polk-DNA-PCNA complex to 3.4 Å resolution and added a low-resolution map (6.4 Å) with Polk with monoubiquitylated PCNA. They have also removed the poorly defined stalled PoID-DNA-PCNA map. The primer extension assays now suggests that the internal PIP-box plays a more important role than the two UBZs motifs in the PCNA-enhanced activity of Polk. These are good changes that have significantly improved the manuscript. However, we still have concerns that need to be addressed.

We thank the reviewer for their comments. The remaining concerns have been addressed as detailed below.

Major concerns:

1) Fig. 6b. The discussion about the "sequential model" is inadequate. The authors have tried and failed to obtain the Polk-PoID-PCNA-DNA super complex either with direct mixing or by SEC column, and 2D EM averages also failed to capture any complex formation. Those results should be presented as supplemental figure and discussed, as they are in support of or at least consistent with the sequential model.

Results of the experiments attempting to observe a Polk-PoID-PCNA-DNA toolbelt are now reported in Supplementary Figure 13, and briefly discussed in the Discussion section. We would like to stress, however, that these are preliminary results which cannot conclusively rule out the toolbelt model.

2) Fig. 6a. A novelty of the work could be Ub interaction with UBZ1. But the experimental support is inadequate. Similarity to RAD18 UBZ interaction with Ub is OK but a bit

handwaving (Fig. 5e). As Polk UBZ1 and UBZ2 are similar in sequence and both could in principle bind to Ub R42 (Fig. 5e). The primer extension assay used combinatorial mutations in both PIP and UBZ1 or UBZ2 and the effects were not clearcut. We will need a clear effect in mutation in UBZ1 but not in UBZ2 to make Fig. 6a convincing. Alternatively, an in vitro binding between UBZ1 domain and Ub but not between UBZ2 and Ub would also work.

Following this reviewer's suggestion, we carried out an in-vitro characterization of Pol k UBZ1 and UBZ2 binding to ubiquitin using NMR. These results are reported in Supplementary Figures 10 and 11. These experiments clearly show that both UBZ1 and UBZ2 bind to a specific site of ubiquitin, in accordance with predictions. UBZ1 appears to bind with a higher affinity than UBZ2, but the limited solubility of the UBZ peptides used prevented a reliable estimation of the binding affinities. Figure 6a has been modified accordingly, showing that both UBZ1 and UBZ2 can in principle bind to ubiquitin attached to PCNA.

A few typos are listed below:

- 1) page 22 line 2, "human" is repetitive.
- 2) page 24 2nd paragraph line 4, the "broth" can be replaced by "media"; the "DNase" throughout the text should be replaced with a formal one "DNase".
- 3) page 25 1st paragraph line 13, missed a preposition after the "eluted".
- 4) page 28 2nd paragraph, "TTP" should be "dTTP", and the formal abbreviation of "K-Ac" should be "KOAc or KAc".
- 5) page 33 4th paragraph line 2, "EMBD" should be "EMDB".
- 6) Page 47 figure legend 5 line 4, there is an extra vertical line following Pol k.
- 7) Fig S6 panel b, missing marker unit "kDa"
- 8) Table S1 1st line 2nd column, "PDD" should be "PDB".

We thank the reviewer for pointing at these typos. Typos have been corrected.

Reviewers' Comments:

Reviewer #2:

Remarks to the Author:

My concerns have been satisfactorily addressed by the authors presenting newl experimental data.

Two minor issues should be taken care of in the final paper: 1) "PAD" should be spelled out in the abstract; 2) When talking about the DNA orientation inside the PCNA bond to Pol Delta, the authors cite only their own published work - papers on the same topic by others should also be cited and briefly commented on.

REVIEWERS' COMMENTS

Reviewer #2 (Remarks to the Author):

My concerns have been satisfactorily addressed by the authors presenting new experimental data.

Two minor issues should be taken care of in the final paper: 1) "PAD" should be spelled out in the abstract; 2) When talking about the DNA orientation inside the PCNA bond to Pol Delta, the authors cite only their own published work - papers on the same topic by others should also be cited and briefly commented on.

We thank the reviewer for their comments. "PAD" has now been spelled out in the abstract. A new paragraph briefly describing previous structures of replicative polymerases bound to PCNA and DNA has been added in the Results section, and the relevant work cited.